# IDENTIFIABLE LATENT POLYNOMIAL CAUSAL MODELS THROUGH THE LENS OF CHANGE

**Yuhang Liu[1], Zhen Zhang[1], Dong Gong[2], Mingming Gong[3,6], Biwei Huang[4]**
**Anton van den Hengel[1], Kun Zhang[5,6], Javen Qinfeng Shi[1]**

[1] Australian Institute for Machine Learning, The University of Adelaide, Australia

[2] School of Computer Science and Engineering, The University of New South Wales, Australia

[3] School of Mathematics and Statistics, The University of Melbourne, Australia

[4] Halicioğlu Data Science Institute (HDSI), University of California San Diego, USA

[5] Department of Philosophy, Carnegie Mellon University, USA

[6] Mohamed bin Zayed University of Artificial Intelligence, United Arab Emirates

`yuhang.liu01@adelaide.edu.au`

## ABSTRACT

Causal representation learning aims to unveil latent high-level causal representations from observed low-level data. One of its primary tasks is to provide reliable assurance of identifying these latent causal models, known as *identifiability*. A recent breakthrough explores identifiability by leveraging the change of causal influences among latent causal variables across multiple environments (Liu et al., 2022). However, this progress rests on the assumption that the causal relationships among latent causal variables adhere strictly to linear Gaussian models. In this paper, we extend the scope of latent causal models to involve nonlinear causal relationships, represented by polynomial models, and general noise distributions conforming to the exponential family. Additionally, we investigate the necessity of imposing changes on all causal parameters and present partial identifiability results when part of them remains unchanged. Further, we propose a novel empirical estimation method, grounded in our theoretical finding, that enables learning consistent latent causal representations. Our experimental results, obtained from both synthetic and real-world data, validate our theoretical contributions concerning identifiability and consistency.

## 1 INTRODUCTION

Causal representation learning, aiming to discover high-level latent causal variables and causal structures among them from unstructured observed data, provides a prospective way to compensate for drawbacks in traditional machine learning paradigms, e.g., the most fundamental limitations that data, driving and promoting the machine learning methods, needs to be independent and identically distributed (i.i.d.) (Schölkopf, 2015). From the perspective of causal representations, the changes in data distribution, arising from various real-world data collection pipelines (Karahan et al., 2016; Frumkin, 2016; Pearl et al., 2016; Chandrasekaran et al., 2021), can be attributed to the changes of causal influences among causal variables (Schölkopf et al., 2021). These changes are observable across a multitude of fields. For instance, these could appear in the analysis of imaging data of cells, where the contexts involve batches of cells exposed to various small-molecule compounds. In this context, each latent variable represents the concentration level of a group of proteins (Chandrasekaran et al., 2021). An inherent challenge with small molecules is their variability in mechanisms of action, which can lead to differences in selectivity (Forbes & Krueger, 2019). In addition, the causal influences of a particular medical treatment on a patient outcome may vary depending on the patient profiles (Pearl et al., 2016). Moreover, causal influences from pollution to health outcomes, such as respiratory illnesses, can vary across different rural environments (Frumkin, 2016).

Despite the above desirable advantages, the fundamental theories underpinning causal representation learning, the issue of identifiability (i.e., uniqueness) of causal representations, remain a significant challenge. One key factor leading to non-identifiability results is that the causal influences among

latent space could be assimilated by the causal influences from latent space to observed space, resulting in multiple feasible solutions (Liu et al., 2022; Adams et al., 2021). To illustrate this, consider two latent causal variables case, and suppose that ground truth is depicted in Figure 1 (a). The causal influence from the latent causal variable $z_1$ to $z_2$ in Figure 1 (a) could be assimilated by the causal influence from $\mathbf{z}$ to $\mathbf{x}$, resulting in the non-identifiability result, as depicted in Figure 1 (b). Efforts to address the transitivity to achieve the identifiability for causal representation learning primarily fall into two categories: 1) enforcing special graph structures (Silva et al., 2006; Cai et al., 2019; Xie et al., 2020; 2022; Adams et al., 2021; Lachapelle et al., 2021), and 2) utilizing the change of causal influences among latent causal variables (Liu et al., 2022; Brehmer et al., 2022; Ahuja et al., 2023; Seigal et al., 2022; Buchholz et al., 2023; Varici et al., 2023). The first approach usually requires special graph structures, i.e., there are two pure child nodes at least for each latent causal variable, as depicted in Figure 1 (c). These pure child nodes essentially prevent the transitivity problem, by the fact that if there is an alternative solution to generate the same observational data, the pure child would not be 'pure' anymore. For example, if the edge from $z_1$ to $z_2$ in Figure 1 (c) is replaced by two new edges (one from $z_1$ to $x_2$, the other from $z_1$ to $x_3$), $x_2$ and $x_3$ are not 'pure' child of $z_2$ anymore. For more details please refer to recent works in Xie et al. (2020; 2022); Huang et al. (2022). However, many causal graphs in reality may be more or less arbitrary, beyond the special graph structures. The second research approach permits any graph structures by utilizing the change of causal influences, as demonstrated in Figure 1 (d). To characterize the change, a surrogate variable $\mathbf{u}$ is introduced into the causal system. Essentially, the success of this approach lies in that the *change* of causal influences in latent space can not be 'absorbed' by the *unchanged* mapping from latent space to observed space across $\mathbf{u}$ (Liu et al., 2022), effectively preventing the transitivity problem. Some methods within this research line require paired interventional data (Brehmer et al., 2022), which may be restricted in some applications such as biology (Stark et al., 2020). Some works require hard interventions or more restricted single-node hard interventions (Ahuja et al., 2023; Seigal et al., 2022; Buchholz et al., 2023; Varici et al., 2023), which could only model specific types of changes. By contrast, the work presented in Liu et al. (2022) studies unpaired data, and employs soft interventions to model a broader range of possible changes, which could be easier to achieve for latent variables than hard interventions.

The work in Liu et al. (2022) compresses the solution space of latent causal variables up to identifiable solutions, particularly *from the perspective of observed data*. This process leverages nonlinear identifiability results from nonlinear ICA (Hyvarinen et al., 2019; Khemakhem et al., 2020; Sorrenson et al., 2020). Most importantly, it relies on some strong assumptions, including 1) causal relations among latent causal variables to be linear Gaussian models, and 2) requiring $\ell + (\ell(\ell+1))/2$ environments where $\ell$ is the number of latent causal variables. By contrast, this work is driven by the realization that we can narrow the solution space of latent causal variables *from the perspective of latent noise variables*, with the identifiability results from nonlinear ICA. This perspective enables us to more effectively utilize model assumptions among latent causal variables, leading to two significant generalizations: 1) Causal relations among latent causal variables can be generalized to be polynomial models with exponential family noise, and 2) The requisite number of environments can be relaxed to $2\ell + 1$ environments, a much more practical number. These two advancements narrow the gap between fundamental theory and practice. Besides, we deeply investigate the assumption of requiring all coefficients within polynomial models to change. We show complete identifiability results if all coefficients change across environments, and partial identifiability results if only part of the coefficients change. The partial identifiability result implies that the whole latent space can be theoretically divided into two subspaces, one relates to invariant latent variables, while the other involves variant variables. This may be potentially valuable for applications that focus on learning invariant latent variables to adapt to varying environments, such as domain adaptation or generalization. To verify our findings, we design a novel method to learn polynomial causal representations in the contexts of Gaussian and non-Gaussian noises. Experiments verify our identifiability results and the efficacy of the proposed approach on synthetic data, image data Ke et al. (2021), and fMRI data.

## 2 RELATED WORK

Due to the challenges of identifiability in causal representation learning, early works focus on learning causal representations in a supervised setting where prior knowledge of the structure of the causal graph of latent variables may be required (Kocaoglu et al., 2018), or additional labels are required to supervise the learning of latent variables (Yang et al., 2021). However, obtaining prior knowledge of

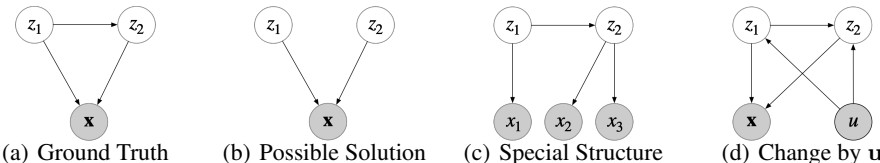

(a) Ground Truth    (b) Possible Solution    (c) Special Structure    (d) Change by **u**

Figure 1: Assume that ground truth is depicted in Figure 1 (a). Due to the transitivity, the graph structure in Figure 1 (b) is an alternative solution for (a), leading to the non-identifiability result. Figure 1 (c) depicts a special structure where two 'pure' child nodes appear. Figure 1 (d) demonstrates the change of the causal influences, characterized by the introduced surrogate variable **u**.

the structure of the latent causal graph is non-trivial in practice, and manual labeling can be costly and error-prone. Some works consider temporal constraint that the effect cannot precede the cause has been used repeatedly in latent causal representation learning (Yao et al., 2021; Lippe et al., 2022; Yao et al., 2022), while this work aims to learn instantaneous causal relations among latent variables. Besides, there are two primary approaches to address the transitivity problem, including imposing special graph structures and using the change of causal influences.

**Special graph structures**   Special graphical structure constraints have been introduced in recent progress in identifiability (Silva et al., 2006; Shimizu et al., 2009; Anandkumar et al., 2013; Frot et al., 2019; Cai et al., 2019; Xie et al., 2020; 2022; Lachapelle et al., 2021). One of representative graph structures is that there are 2 pure children for each latent causal variables (Xie et al., 2020; 2022; Huang et al., 2022). These special graph structures are highly related to sparsity, which implies that a sparser model that fits the observation is preferred (Adams et al., 2021). However, many latent causal graphs in reality may be more or less arbitrary, beyond a purely sparse graph structure. Differing from special graph structures, this work does not restrict graph structures among latent causal variables, by exploring the change of causal influence among latent causal variables.

**The change of causal influence**   Very recently, there have been some works exploring the change of causal influence (Von Kügelgen et al., 2021; Liu et al., 2022; Brehmer et al., 2022; Ahuja et al., 2023; Seigal et al., 2022; Buchholz et al., 2023; Varici et al., 2023). Roughly speaking, these changes of causal influences could be categorized as hard interventions or soft interventions. Most of them consider hard intervention or more restricted single-node hard interventions (Ahuja et al., 2023; Seigal et al., 2022; Buchholz et al., 2023; Varici et al., 2023), which can only capture some special changes of causal influences. In contrast, soft interventions could model more possible types of change (Liu et al., 2022; Von Kügelgen et al., 2021), which could be easier to achieve in latent space. Differing from the work in Von Kügelgen et al. (2021) that identifies two coarse-grained latent subspaces, e.g., style and content, the work in Liu et al. (2022) aims to identify fine-grained latent variables. In this work, we generalize the identifiability results in Liu et al. (2022), and relax the requirement of the number of environments. Moreover, we discuss the necessity of requiring all causal influences to change, and partial identifiability results when part of causal influences changes.

# 3   IDENTIFIABLE CAUSAL REPRESENTATIONS WITH VARYING POLYNOMIAL CAUSAL MODELS

In this section, we show that by leveraging changes, latent causal representations are identifiable (including both latent causal variables and the causal model), for general nonlinear models and noise distributions are sampled from two-parameter exponential family members. Specifically, we start by introducing our defined varying latent polynomial causal models in Section 3.1, aiming to facilitate comprehension of the problem setting and highlight our contributions. Following this, in Section 3.2, we present our identifiability results under the varying latent polynomial causal model. This constitutes a substantial extension beyond previous findings within the domain of varying linear Gaussian models (Liu et al., 2022). Furthermore, we delve into a thorough discussion about the necessity of requiring changes in all causal influences among the latent causal variables. We, additionally, show partial identifiability results in cases where only a subset of causal influences changes in Section 3.3, further solidifying our identifiability findings.

## 3.1 Varying Latent Polynomial Causal Models

We explore causal generative models where the observed data $\mathbf{x}$ is generated by the latent causal variables $\mathbf{z} \in \mathbb{R}^{\ell}$, allowing for any potential graph structures among $\mathbf{z}$. In addition, there exist latent noise variables $\mathbf{n} \in \mathbb{R}^{\ell}$, known as exogenous variables in causal systems, corresponding to latent causal variables. We introduce a surrogate variable $\mathbf{u}$ characterizing the changes in the distribution of $\mathbf{n}$, as well as the causal influences among latent causal variables $\mathbf{z}$. Here $\mathbf{u}$ could be environment, domain, or time index. More specifically, we parameterize the causal generative models by assuming $\mathbf{n}$ follows an exponential family given $\mathbf{u}$, and assuming $\mathbf{z}$ and $\mathbf{x}$ are generated as follows:

$$p_{(\mathbf{T},\boldsymbol{\eta})}(\mathbf{n}|\mathbf{u}) := \prod_i \frac{1}{Z_i(\mathbf{u})} \exp[\sum_j (T_{i,j}(n_i)\eta_{i,j}(\mathbf{u}))], \tag{1}$$

$$z_i := \mathrm{g}_i(\mathrm{pa}_i, \mathbf{u}) + n_i, \tag{2}$$

$$\mathbf{x} := \mathbf{f}(\mathbf{z}) + \boldsymbol{\varepsilon}, \tag{3}$$

with

$$\mathrm{g}_i(\mathbf{z}, \mathbf{u}) = \boldsymbol{\lambda}_i^T(\mathbf{u})[\mathbf{z}, \mathbf{z}\bar{\otimes}\mathbf{z}, ..., \mathbf{z}\bar{\otimes}...\bar{\otimes}\mathbf{z}], \tag{4}$$

where

- in Eq. 1, $Z_i(\mathbf{u})$ denotes the normalizing constant, and $T_{i,j}(n_i)$ denotes the sufficient statistic for $n_i$, whose the natural parameter $\eta_{i,j}(\mathbf{u})$ depends on $\mathbf{u}$. Here we focus on two-parameter (e.g., $j \in \{1, 2\}$) exponential family members, which include not only Gaussian, but also inverse Gaussian, Gamma, inverse Gamma, and beta distributions.

- In Eq. 2, $\mathrm{pa}_i$ denotes the set of parents of $z_i$.

- In Eq. 3, $\mathbf{f}$ denotes a nonlinear mapping, and $\boldsymbol{\varepsilon}$ is independent noise with probability density function $p_{\boldsymbol{\varepsilon}}(\boldsymbol{\varepsilon})$.

- In Eq. 4, where $\boldsymbol{\lambda_i}(\mathbf{u}) = [\lambda_{1,i}(\mathbf{u}), \lambda_{2,i}(\mathbf{u}), ...]$, $\bar{\otimes}$ represents the Kronecker product with all distinct entries, e.g., for 2-dimension case, $z_1\bar{\otimes}z_2 = [z_1^2, z_2^2, z_1 z_2]$. Here $\boldsymbol{\lambda_i}(\mathbf{u})$ adheres to common Directed Acyclic Graphs (DAG) constraints.

The models defined above represent polynomial models and two-parameter exponential family distributions, which include not only Gaussian, but also inverse Gaussian, Gamma, inverse Gamma, and beta distributions. Clearly, the linear Gaussian models proposed in Liu et al. (2022) can be seen as a special case in this broader framework. The proposed latent causal models, as defined in Eqs. 1 - 4, have the capacity to capture a wide range of change of causal influences among latent causal variables, including a diverse set of nonlinear functions and two-parameter exponential family noises. This expansion in scope serves to significantly bridge the divide between foundational theory and practical applications.

## 3.2 Complete Identifyability Result

The crux of our identifiability analysis lies in leveraging the changes in causal influences among latent causal variables, orchestrated by $\mathbf{u}$. Unlike many prior studies that constrain the changes within the specific context of hard interventions (Brehmer et al., 2022; Ahuja et al., 2023; Seigal et al., 2022; Buchholz et al., 2023; Varici et al., 2023), our approach welcomes and encourages changes. Indeed, our approach allows a wider range of potential changes which can be interpreted as soft interventions (via the causal generative model defined in Eqs. 1- 4).

**Theorem 3.1** *Suppose latent causal variables $\mathbf{z}$ and the observed variable $\mathbf{x}$ follow the causal generative models defined in Eq. 1 - Eq. 4. Assume the following holds:*

- *(i) The set $\{\mathbf{x} \in \mathcal{X} | \varphi_{\varepsilon}(\mathbf{x}) = 0\}$ has measure zero where $\varphi_{\boldsymbol{\varepsilon}}$ is the characteristic function of the density $p_{\boldsymbol{\varepsilon}}$,*

- *(ii) The function $\mathbf{f}$ in Eq. 3 is bijective,*

*(iii) There exist $2\ell + 1$ values of $\mathbf{u}$, i.e., $\mathbf{u}_0, \mathbf{u}_1, ..., \mathbf{u}_{2\ell}$, such that the matrix*

$$\mathbf{L} = (\boldsymbol{\eta}(\mathbf{u} = \mathbf{u}_1) - \boldsymbol{\eta}(\mathbf{u} = \mathbf{u}_0), ..., \boldsymbol{\eta}(\mathbf{u} = \mathbf{u}_{2\ell}) - \boldsymbol{\eta}(\mathbf{u} = \mathbf{u}_0)) \tag{5}$$

*of size $2\ell \times 2\ell$ is invertible. Here $\boldsymbol{\eta}(\mathbf{u}) = [\eta_{i,j}(\mathbf{u})]_{i,j}$,*

*(iv) The function class of $\lambda_{i,j}$ can be expressed by a Taylor series: for each $\lambda_{i,j}$, $\lambda_{i,j}(\mathbf{u} = \mathbf{0}) = 0$,*

*then the true latent causal variables $\mathbf{z}$ are related to the estimated latent causal variables $\hat{\mathbf{z}}$, which are learned by matching the true marginal data distribution $p(\mathbf{x}|\mathbf{u})$, by the following relationship: $\mathbf{z} = \mathbf{P}\hat{\mathbf{z}} + \mathbf{c}$, where $\mathbf{P}$ denotes the permutation matrix with scaling, $\mathbf{c}$ denotes a constant vector.*

**Proof sketch** First, we demonstrate that given the DAG (Directed Acyclic Graphs) constraint and the assumption of additive noise in latent causal models as in Eq. 4, the identifiability result in Sorrenson et al. (2020) holds. Specifically, it allows us to identify the latent noise variables $\mathbf{n}$ up to scaling and permutation, e.g., $\mathbf{n} = \mathbf{P}\hat{\mathbf{n}} + \mathbf{c}$ where $\hat{\mathbf{n}}$ denotes the recovered latent noise variables obtained by matching the true marginal data distribution. Building upon this result, we then leverage polynomial property that the composition of polynomials is a polynomial, and additive noise assumption defined in Eq. 2, to show that the latent causal variables $\mathbf{z}$ can also be identified up to polynomial transformation, i.e., $\mathbf{z} = Poly(\hat{\mathbf{z}}) + \mathbf{c}$ where $Poly$ denotes a polynomial function. Finally, using the change of causal influences among $\mathbf{z}$, the polynomial transformation can be further reduced to permutation and scaling, i.e., $\mathbf{z} = \mathbf{P}\hat{\mathbf{z}} + \mathbf{c}$. Detailed proof can be found in A.2.

Our model assumptions among latent causal variables is a typical additive noise model as in Eq. 4. Given this, the identifiability of latent causal variables implies that the causal graph is also identified. This arises from the fact that additive noise models are identifiable (Hoyer et al., 2008; Peters et al., 2014), regardless of the scaling on $\mathbf{z}$. In addition, the proposed identifiability result in Theorem 3.1 represents a more generalized form of the previous result in Liu et al. (2022). When the polynomial model's degree is set to 1 and the noise is sampled from Gaussian distribution, the proposed identifiability result in Theorem 3.1 converges to the earlier finding in Liu et al. (2022). Notably, the proposed identifiability result requires only $2\ell + 1$ environments, while the result in Liu et al. (2022) needs the number of environments depending on the graph structure among latent causal variables. In the worst case, e.g., a full-connected causal graph over latent causal variables, i.e., $\ell + (\ell(\ell + 1))/2$.

### 3.3 COMPLETE AND PARTIAL CHANGE OF CAUSAL INFLUENCES

The aforementioned theoretical result necessitates that all coefficients undergo changes across various environments, as defined in Eq. 4. However, in practical applications, the assumption may not hold true. Consequently, two fundamental questions naturally arise: is the assumption necessary for identifiability, in the absence of any supplementary assumptions? Alternatively, can we obtain partial identifiability results if only part of the coefficients changes across environments? In this section, we provide answers to these two questions.

**Corollary 3.2** *Suppose latent causal variables $\mathbf{z}$ and the observed variable $\mathbf{x}$ follow the causal generative models defined in Eq. 1 - Eq. 4. Under the condition that the assumptions (i)-(iii) in Theorem 3.1 are satisfied, if there is an unchanged coefficient in Eq. 4 across environments, $\mathbf{z}$ is unidentifiable, without additional assumptions.*

**Proof sketch** The proof of the above corollary can be done by investigating whether we can always construct an alternative solution, different from $\mathbf{z}$, to generate the same observation $\mathbf{x}$, if there is an unchanged coefficient across $\mathbf{u}$. The construction can be done by the following: assume that there is an unchanged coefficient in polynomial for $z_i$, we can always construct an alternative solution $\mathbf{z}'$ by removing the term involving the unchanged coefficient in polynomial $g_i$, while keeping the other unchanged, i.e., $z_j' = z_j$ for all $j \neq i$. Details can be found in A.3.

**Insights** 1) This corollary implies the necessity of requiring all coefficients to change to obtain the complete identifyability result, without introducing additional assumptions. We acknowledge the possibility of mitigating this requirement by imposing specific graph structures, which is beyond

the scope of this work. However, it is interesting to explore the connection between the change of causal influences and special graph structures for the identifiability of causal representations in the future. 2) In addition, this necessity may depend on specific model assumptions. For instance, if we use MLPs to model the causal relations of latent causal variables, it may be not necessary to require all weights in the MLPs to change.

Requiring all coefficients to change might be challenging in real applications. In fact, when part of the coefficients change, we can still provide partial identifiability results, as outlined below:

**Corollary 3.3** *Suppose latent causal variables* $\mathbf{z}$ *and the observed variable* $\mathbf{x}$ *follow the causal generative models defined in Eq. 1 - Eq. 4. Under the condition that the assumptions (i)-(iii) in Theorem 3.1 are satisfied, for each* $z_i$,

*(a) if it is a root node or all coefficients in the corresponding polynomial* $g_i$ *change in Eq. 4, then the true* $z_i$ *is related to the recovered one* $\hat{z}_j$, *obtained by matching the true marginal data distribution* $p(\mathbf{x}|\mathbf{u})$, *by the following relationship:* $z_i = s\hat{z}_j + c$, *where* $s$ *denotes scaling,* $c$ *denotes a constant,*

*(b) if there exists an unchanged coefficient in polynomial* $g_i$ *in Eq. 4, then* $z_i$ *is unidentifiable.*

**Proof sketch** This can be proved by the fact that regardless of the change of the coefficients, two results hold, i.e., $\mathbf{z} = Poly(\hat{\mathbf{z}}) + \mathbf{c}$, and $\mathbf{n} = \mathbf{P}\hat{\mathbf{n}} + \mathbf{c}$. Then using the change of all coefficients in the corresponding polynomial $g_i$, we can prove (a). For (b), similar to the proof of corollary 3.2, we can construct a possible solution $z'_i$ for $z_i$ by removing the term corresponding to the unchanged coefficient, resulting in an unidentifiable result.

**Insights** 1) The aforementioned partial identifiability result implies that the entire latent space can theoretically be partitioned into two distinct subspaces: one subspace pertains to invariant latent variables, while the other encompasses variant variables. This may be potentially valuable for applications that focus on learning invariant latent variables to adapt to varying environments, such as domain adaptation (or generalization) (Liu et al., 2024). 2) In cases where there exists an unchanged coefficient in the corresponding polynomial $g_i$, although $z_i$ is not entirely identifiable, we may still ascertain a portion of $z_i$. To illustrate this point, for simplicity, assume that $z_2 = 3z_1 + \lambda_{1,2}(\mathbf{u})z_1^2 + n_2$. Our result (b) shows that $z_2$ is unidentifiable due to the constant coefficient 3 on the right side of the equation. However, the component $\lambda_{1,2}(\mathbf{u})z_1^2 + n_2$ may still be identifiable. While we refrain from presenting a formal proof for this insight in this context, we can provide some elucidation. If we consider $z_2$ as a composite of two variables, $z_a = 3z_1$ and $z_b = \lambda_{1,2}(\mathbf{u})z_1^2 + n_2$, according to our finding (a), $z_b$ may be identified.

## 4 LEARNING POLYNOMIAL CAUSAL REPRESENTATIONS

In this section, we translate our theoretical findings into a novel algorithm. Following the work in Liu et al. (2022), due to permutation indeterminacy in latent space, we can naturally enforce a causal order $z_1 \succ z_2 \succ ..., \succ z_\ell$ to impose each variable to learn the corresponding latent variables in the correct causal order. As a result, for the Gaussian noise, in which the conditional distributions $p(z_i|pa_i)$, where $pa_i$ denote the parent nodes of $z_i$, can be expressed as an analytic form, we formulate the prior distribution as conditional Gaussian distributions. Differing from the Gaussian noise, non-Gaussian noise does not have an analytically tractable solution, in general. Given this, we model the prior distribution of $p(\mathbf{z}|\mathbf{u})$ by $p(\boldsymbol{\lambda}, \mathbf{n}|\mathbf{u})$. As a result, we arrive at:

$$p(\mathbf{z}|\mathbf{u}) = \begin{cases} p(z_1|\mathbf{u}) \prod_{i=2}^{\ell} p(z_i|\mathbf{z}_{<i}, \mathbf{u}) = \prod_{i=1}^{\ell} \mathcal{N}(\mu_{z_i}(\mathbf{u}), \delta_{z_i}^2(\mathbf{u})), & \text{if } \mathbf{n} \sim \text{Gaussian} \\ \left( \prod_{i=1}^{\ell} \prod_{j=1}^{\ell} p(\lambda_{j,i}|\mathbf{u}) \right) \prod_{i=1}^{\ell} p(n_i|\mathbf{u}), & \text{if } \mathbf{n} \sim \text{non-Gaussian} \end{cases} \tag{6}$$

where $\mathcal{N}(\mu_{z_i}(\mathbf{u}), \delta_{z_i}^2(\mathbf{u}))$ denotes the Gaussian probability density function with mean $\mu_{z_i}(\mathbf{u})$ and variance $\delta_{z_i}^2(\mathbf{u})$. Note that non-Gaussian noises typically tend to result in high-variance gradients. They often require distribution-specific variance reduction techniques to be practical, which is beyond the scope of this paper. Instead, we straightforwardly use the PyTorch (Paszke et al., 2017)

implementation of the method of Jankowiak & Obermeyer (2018), which computes implicit reparameterization using a closed-form approximation of the probability density function derivative. In our implementation, we found that the implicit reparameterization leads to high-variance gradients for inverse Gaussian and inverse Gamma distributions. Therefore, we present the results of Gamma, and beta distributions in experiments.

**Prior on coefficients** $p(\lambda_{j,i})$    We enforce two constraints on the coefficients, DAG constraint and sparsity. The DAG constraint is to ensure a directed acyclic graph estimation. Current methods usually employ a relaxed DAG constraint proposed by Zheng et al. (2018) to estimate causal graphs, which may result in a cyclic graph estimation due to the inappropriate setting of the regularization hyperparameter. Following the work in Liu et al. (2022), we can naturally ensure a directed acyclic graph estimation by enforcing the coefficients matrix $\boldsymbol{\lambda}(\mathbf{u})^T$ to be a lower triangular matrix corresponding to a fully-connected graph structure, due to permutation property in latent space. In addition, to prune the fully connected graph structure to select true parent nodes, we enforce a sparsity constraint on each $\lambda_{j,i}(\mathbf{u})$. In our implementation, we simply impose a Laplace distribution on each $\lambda_{j,i}(\mathbf{u})$, other distributions may also be flexible, e.g, horseshoe prior (Carvalho et al., 2009) or Gaussian prior with zero mean and variance sampled from a uniform prior (Liu et al., 2019).

**Variational Posterior**    We employ variational posterior to approximate the true intractable posterior of $p(\mathbf{z}|\mathbf{x}, \mathbf{u})$. The nature of the proposed prior in Eq. 6 gives rise to the posterior:

$$q(\mathbf{z}|\mathbf{u}, \mathbf{x}) = \begin{cases} q(z_1|\mathbf{u}, \mathbf{x}) \prod_{i=2}^{\ell} q(z_i|\mathbf{z}_{<i}, \mathbf{u}, \mathbf{x}), & \text{if } \mathbf{n} \sim \text{Gaussian} \\ \left( \prod_{i=1}^{\ell} \prod_{j=1} q(\lambda_{j,i}|\mathbf{x}, \mathbf{u}) \right) \prod_{i=1}^{\ell} q(n_i|\mathbf{u}, \mathbf{x}), & \text{if } \mathbf{n} \sim \text{non-Gaussian} \end{cases} \quad (7)$$

where variational posteriors $q(z_i|\mathbf{z}_{<i}, \mathbf{u}, \mathbf{x})$, $q(\lambda_{j,i})$ and $q(n_i|\mathbf{u})$ employ the same distribution as their priors, so that an analytic form of Kullback–Leibler divergence between the variational posterior and the prior can be provided. As a result, we can arrive at a simple objective:

$$\max \mathbb{E}_{q(\mathbf{z}|\mathbf{x},\mathbf{u})q(\boldsymbol{\lambda}|\mathbf{x},\mathbf{u})}(p(\mathbf{x}|\mathbf{z}, \mathbf{u})) - D_{KL}(q(\mathbf{z}|\mathbf{x},\mathbf{u})||p(\mathbf{z}|\mathbf{u})) - D_{KL}(q(\boldsymbol{\lambda}|\mathbf{x},\mathbf{u})||p(\boldsymbol{\lambda}|\mathbf{u})), \quad (8)$$

where $D_{KL}$ denotes the KL divergence. Implementation details can be found in Appendix A.6.

## 5    EXPERIMENTS

**Synthetic Data**    We first conduct experiments on synthetic data, generated by the following process: we divide latent noise variables into $M$ segments, where each segment corresponds to one value of $\mathbf{u}$ as the segment label. Within each segment, the location and scale parameters are respectively sampled from uniform priors. After generating latent noise variables, we randomly generate coefficients for polynomial models, and finally obtain the observed data samples by an invertible nonlinear mapping on the polynomial models. More details can be found in Appendix A.5.

We compare the proposed method with vanilla VAE (Kingma & Welling, 2013), $\beta$-VAE (Higgins et al., 2017), identifiable VAE (iVAE) (Khemakhem et al., 2020). Among them, iVAE is able to identify the true independent noise variables up to permutation and scaling, with certain assumptions. $\beta$-VAE has been widely used in various disentanglement tasks, motivated by enforcing independence among the recovered variables, but it has no theoretical support. Note that both methods assume that the latent variables are independent, and thus they cannot model the relationships among latent variables. All these methods are implemented in three different settings corresponding to linear models with Beta distributions, linear models with Gamma noise, and polynomial models with Gaussian noise, respectively. To make a fair comparison, for non-Gaussian noise, all these methods use the PyTorch (Paszke et al., 2017) implementation of the method of Jankowiak & Obermeyer (2018) to compute implicit reparameterization. We compute the mean of the Pearson correlation coefficient (MPC) to evaluate the performance of our proposed method. Further, we report the structural Hamming distance (SHD) of the recovered latent causal graphs by the proposed method.

Figure 2 shows the performances of different methods on different models, in the setting where all coefficients change across $\mathbf{u}$. According to MPC, the proposed method with different model assumptions obtains satisfactory performance, which verifies the proposed identifiability results. Further, Figure 3 shows the performances of the proposed method when part of coefficients change across $\mathbf{u}$, for which we can see that unchanged weight leads to non-identifiability results, and changing

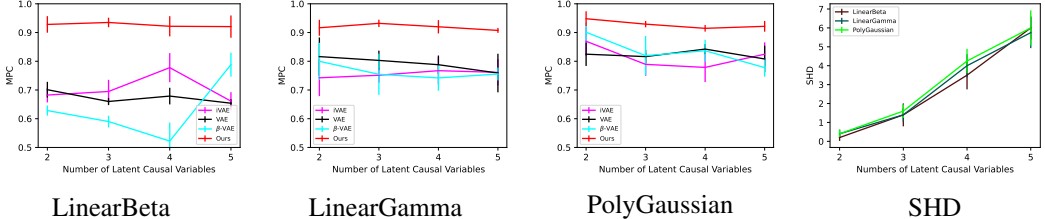

Figure 2: Performances of different methods on linear models with Beta noise, linear models with Gamma noise, and polynomial models with Gaussian noises. In terms of MPC, the proposed method performs better than others, which verifies the proposed identifiablity results. The right subfigure shows the SHD obtained by the proposed method in different model assumptions.

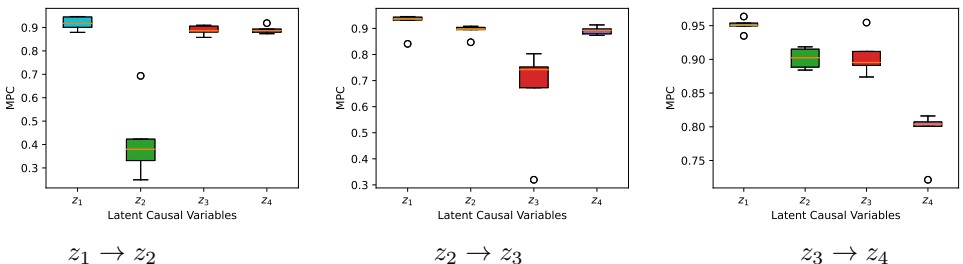

Figure 3: Performances of the proposed method with the change of part of weights, on linear models with Beta noise. The ground truth of the causal graph is $z_1 \to z_2 \to z_3 \to z_4$. From left to right: keeping weight on $z_1 \to z_2$, $z_2 \to z_3$, and $z_3 \to z_4$ unchanged. Those results are consistent with the analysis of partial identifiability results in corollary 3.3.

weights contribute to the identifiability of the corresponding nodes. These empirical results are consistent with partial identifiability results in corollary 3.3.

**Image Data** We further verify the proposed identifiability results and method on images from the chemistry dataset proposed in Ke et al. (2021), which corresponds to chemical reactions where the state of an element can cause changes to another variable's state. The images consist of a number of objects whose positions are kept fixed, while the colors (states) of the objects change according to the causal graph. To meet our assumptions, we use a weight-variant linear causal model with Gamma noise to generate latent variables corresponding to the colors. The ground truth of the latent causal graph is that the 'diamond' (e.g., $z_1$) causes the 'triangle' (e.g., $z_2$), and the 'triangle' causes the 'square' (e.g., $z_3$). A visualization of the observational images can be found in Figure 4. Figure 5 shows the MPC obtained by different methods. The proposed method performs better than others. The proposed method also learns the correct causal graph as verified by intervention results in Figure 6, i.e., 1) intervention on $z_1$ ('diamond') causes the change of both $z_2$ ('triangle') and $z_3$ ('square'), 2) intervention on $z_2$ only causes the change of $z_3$, 3) intervention on $z_3$ can not affect both $z_1$ and $z_2$. These results are consistent with the correct causal graph, i.e., $z_1 \to z_2 \to z_3$. Due to limited space, more traversal results on the learned latent variables by the other methods can be found in Appendix A.7. For these methods, since there is no identifiability guarantee, we found that traversing on each learned variable leads to the colors of all objects changing.

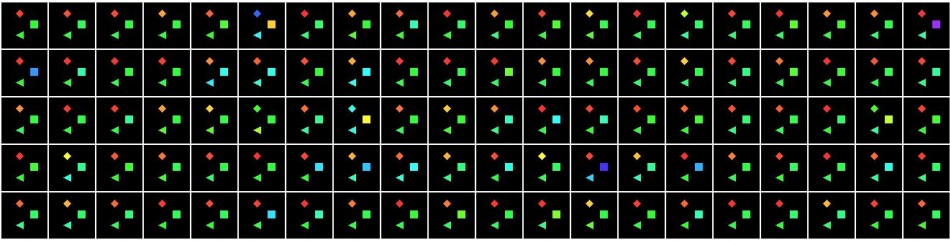

Figure 4: Samples from the image dataset generated by modifying the chemistry dataset in Ke et al. (2021). The colors (states) of the objects change according to the causal graph: the 'diamond' causes the 'triangle', and the 'triangle' causes the 'square', i.e., $z_1 \to z_2 \to z_3$.

|  | $z_1$ | $z_2$ | $z_3$ |
|---|---|---|---|
| $\hat{z}_1$ | 0.963 | 0.005 | 0.115 |
| $\hat{z}_2$ | 0.008 | 0.702 | 0.491 |
| $\hat{z}_3$ | 0.085 | 0.421 | 0.803 |

iVAE

|  | $z_1$ | $z_2$ | $z_3$ |
|---|---|---|---|
| $\hat{z}_1$ | 0.067 | 0.582 | 0.628 |
| $\hat{z}_2$ | 0.958 | 0.065 | 0.046 |
| $\hat{z}_3$ | 0.117 | 0.429 | 0.765 |

$\beta$-VAE

|  | $z_1$ | $z_2$ | $z_3$ |
|---|---|---|---|
| $\hat{z}_1$ | 0.053 | 0.742 | 0.228 |
| $\hat{z}_2$ | 0.858 | 0.265 | 0.046 |
| $\hat{z}_3$ | 0.117 | 0.429 | 0.765 |

VAE

|  | $z_1$ | $z_2$ | $z_3$ |
|---|---|---|---|
| $\hat{z}_1$ | 0.953 | 0.035 | 0.068 |
| $\hat{z}_2$ | 0.137 | 0.852 | 0.392 |
| $\hat{z}_3$ | 0.186 | 0.401 | 0.871 |

Ours

Figure 5: MPC obtained by different methods on the image dataset, the proposed method performs better than others, supported by our identifiability.

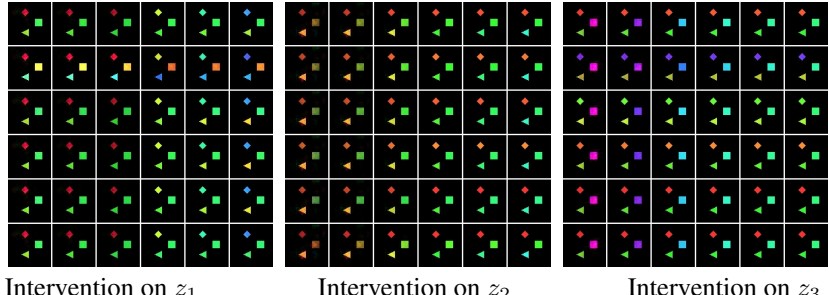

Intervention on $z_1$          Intervention on $z_2$          Intervention on $z_3$

Figure 6: Intervention results obtained by the proposed method on the image data. From left to right: interventions on the learned $z_1, z_2, z_3$, respectively. The vertical axis denotes different samples, The horizontal axis denotes enforcing different values on the learned causal representation.

**fMRI Data**  Following Liu et al. (2022), we further apply the proposed method to fMRI hippocampus dataset (Laumann & Poldrack, 2015), which contains signals from six separate brain regions: perirhinal cortex (PRC), parahippocampal cortex (PHC), entorhinal cortex (ERC), subiculum (Sub), CA1, and CA3/Dentate Gyrus (DG). These signals were recorded during resting states over a span of 84 consecutive days from the same individual. Each day's data is considered a distinct instance, resulting in an 84-dimensional vector represented as **u**. Given our primary interest in uncovering latent causal variables, we treat the six signals as latent causal variables. To transform them into observed data, we subject them to a random nonlinear mapping. Subsequently, we apply our proposed method to the transformed observed data. We then apply the proposed method to the transformed observed data to recover the latent causal variables. Figure 7 shows the results obtained by the proposed method with different model assumptions. We can see that the polynomial models with Gaussian noise perform better than others, and the result obtained by linear models with Gaussian noise is suboptimal. This also may imply that 1) Gaussian distribution is more reasonable to model the noise in this data, 2) linear relations among these signals may be more dominant than nonlinear.

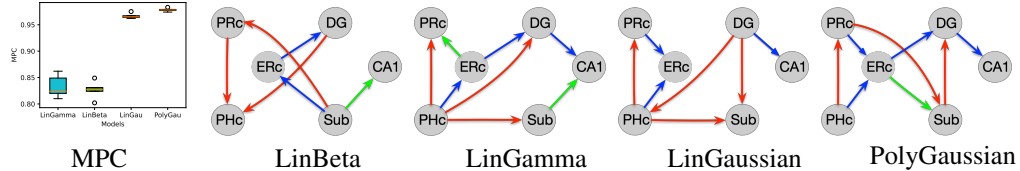

MPC          LinBeta          LinGamma          LinGaussian          PolyGaussian

Figure 7: MPC obtained by the proposed method with different noise assumptions. Blue edges are feasible given anatomical connectivity, red edges are not, and green edges are reversed.

## 6 CONCLUSION

Identifying latent causal representations is known to be generally impossible without certain assumptions. This work generalizes the previous linear Gaussian models to polynomial models with two-parameter exponential family members, including Gaussian, inverse Gaussian, Gamma, inverse Gamma, and Beta distribution. We further discuss the necessity of requiring all coefficients in polynomial models to change to obtain complete identifiability result, and analyze partial identifiability results in the setting where only part of the coefficients change. We then propose a novel method to learn polynomial causal representations with Gaussian or non-Gaussian noise. Experimental results on synthetic and real data demonstrate our findings and consistent results. Identifying causal representations by exploring the change of causal influences is still an open research line. In addition, even with the identifiability guarantees, it is still challenging to learn causal graphs in latent space.

## 7 ACKNOWLEDGEMENTS

We are very grateful to the anonymous reviewers for their help in improving the paper. YH was partially supported by Centre for Augmented Reasoning. DG was partially supported by an ARC DECRA Fellowship DE230101591. MG was supported by ARC DE210101624. KZ would like to acknowledge the support from NSF Grant 2229881, the National Institutes of Health (NIH) under Contract R01HL159805, and grants from Apple Inc., KDDI Research Inc., Quris AI, and Infinite Brain Technology.

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

# A APPENDIX

## A.1 THE RESULT IN (LIU ET AL., 2022)

For comparison, here we provide the main model assumptions and result in the work by (Liu et al., 2022). It considers the following causal generative models:

$$n_i :\sim \mathcal{N}(\eta_{i,1}(\mathbf{u}), \eta_{i,2}(\mathbf{u})), \tag{9}$$

$$z_i := \boldsymbol{\lambda}_i^T(\mathbf{u})(\mathbf{z}) + n_i, \tag{10}$$

$$\mathbf{x} := \mathbf{f}(\mathbf{z}) + \boldsymbol{\varepsilon} \tag{11}$$

**Theorem A.1** *Suppose latent causal variables* $\mathbf{z}$ *and the observed variable* $\mathbf{x}$ *follow the generative models defined in Eq. 9- Eq. 11, with parameters* $(\mathbf{f}, \boldsymbol{\lambda}, \boldsymbol{\eta})$. *Assume the following holds:*

*(i) The set* $\{\mathbf{x} \in \mathcal{X} | \varphi_{\varepsilon}(\mathbf{x}) = 0\}$ *has measure zero (i.e., has at the most countable number of elements), where* $\varphi_{\boldsymbol{\varepsilon}}$ *is the characteristic function of the density* $p_{\boldsymbol{\varepsilon}}$.

*(ii) The function* $\mathbf{f}$ *in Eq. 11 is bijective.*

*(iii) There exist* $2\ell + 1$ *distinct points* $\mathbf{u_{n,0}}, \mathbf{u_{n,1}}, ..., \mathbf{u_{n,2\ell}}$ *such that the matrix*

$$\mathbf{L_n} = (\boldsymbol{\eta}(\mathbf{u_{n,1}}) - \boldsymbol{\eta}(\mathbf{u_{n,0}}), ..., \boldsymbol{\eta}(\mathbf{u_{n,2\ell}}) - \boldsymbol{\eta}(\mathbf{u_{n,0}})) \tag{12}$$

*of size* $2\ell \times 2\ell$ *is invertible.*

*(iv) There exist* $k + 1$ *distinct points* $\mathbf{u_{z,0}}, \mathbf{u_{z,1}}, ..., \mathbf{u_{z,k}}$ *such that the matrix*

$$\mathbf{L_z} = (\boldsymbol{\eta_z}(\mathbf{u_{z,1}}) - \boldsymbol{\eta_z}(\mathbf{u_{z,0}}), ..., \boldsymbol{\eta_z}(\mathbf{u_{z,k}}) - \boldsymbol{\eta_z}(\mathbf{u_{z,0}})) \tag{13}$$

*of size* $k \times k$ *is invertible.*

*(v) The function class of* $\lambda_{i,j}$ *can be expressed by a Taylor series: for each* $\lambda_{i,j}$, $\lambda_{i,j}(\mathbf{0}) = 0$,

*then the recovered latent causal variables* $\hat{\mathbf{z}}$, *which are learned by matching the true marginal data distribution* $p(x|u)$, *are related to the true latent causal variables* $\mathbf{z}$ *by the following relationship:* $\mathbf{z} = \mathbf{P}\hat{\mathbf{z}} + \mathbf{c}$, *where* $\mathbf{P}$ *denotes the permutation matrix with scaling,* $\mathbf{c}$ *denotes a constant vector.*

Here $\boldsymbol{\eta}$ denote sufficient statistic of distribution of latent noise variables $\mathbf{n}$, $\boldsymbol{\eta_z}$ denote sufficient statistic of distribution of latent causal variables $\mathbf{z}$. $k$ denotes the number of the sufficient statistic of $\mathbf{z}$. Please refer to (Liu et al., 2022) for more details.

Compared with the work in (Liu et al., 2022), this work generalizes linear Gaussian models in Eq. 10 to polynomial models with two-parameter exponential family, as defined in Eq. 1-2. In addition, this work removes the assumption (iv), which requires the number of environments highly depending on the graph structure. Moreover, both the work in (Liu et al., 2022) and this work explores the change of causal influences, in this work, we provide analysis for the necessity of requiring all causal influence to change, and also partial identifiability results when part of causal influences changes. This analysis enables the research line, allowing causal influences to change, more solid.

A.2 THE PROOF OF THEOREM 3.1

For convenience, we first introduce the following lemmas.

**Lemma A.2** $\mathbf{z}$ *can be expressed as a polynomial function with respect to* $\mathbf{n}$, *i.e.,* $\mathbf{z} = \mathbf{h}(\mathbf{n}, \mathbf{u})$, *where* $\mathbf{h}$ *denote a polynomial, and* $\mathbf{h}^{-1}$ *is also a polynomial function.*

Proof can be easily shown by the following: since we have established that $z_i$ depends on its parents and $n_i$ as defined in Eqs. 2 and 4, we can recursively express $z_i$ in terms of latent noise variables relating to its parents and $n_i$ using the equations provided in Eqs. 2 and 4. Specifically, without loss of the generality, suppose that the correct causal order is $z_1 \succ z_2 \succ ... \succ z_\ell$, we have:

$$z_1 = \underbrace{n_1}_{h_1(n_1)},$$

$$z_2 = g_2(z_1) + n_2 = \underbrace{g_2(n_1, \mathbf{u}) + n_2}_{h_2(n_1, n_2, \mathbf{u})},$$

$$z_3 = \underbrace{g_3(z_1, g_2(n_1, \mathbf{u}) + n_2, \mathbf{u}) + n_3}_{h_3(n_1, n_2, n_3, \mathbf{u})}, \tag{14}$$

$$......,$$

where $\mathbf{h}(\mathbf{n}, \mathbf{u}) = [h_1(n_1, \mathbf{u}), h_2(n_1, n_2, \mathbf{u}), h_3(n_1, n_2, n_3, \mathbf{u})...]$. By the fact that the composition of polynomials is still a polynomial, and repeating the above process for each $z_i$ can show that $\mathbf{z}$ can be expressed as a polynomial function with respect to $\mathbf{n}$, i.e., $\mathbf{z} = \mathbf{h}(\mathbf{n}, \mathbf{u})$. Further, according to the additive noise models and DAG constraints, it can be shown that the Jacobi determinant of $\mathbf{h}$ equals 1, and thus the mapping $\mathbf{h}$ is invertible. Moreover, $\mathbf{h}^{-1}$ can be recursively expressed in terms of $z_i$ according to Eq. 14, as follows:

$$n_1 = \underbrace{z_1}_{h_1^{-1}(n_1)},$$

$$n_2 = z_2 - g_2(n_1, \mathbf{u}) = \underbrace{z_2 - g_2(z_1, \mathbf{u})}_{h_2^{-1}(z_1, z_2, \mathbf{u})},$$

$$n_3 = z_3 - g_3(z_1, g_2(n_1, \mathbf{u}) + n_2, \mathbf{u}) = \underbrace{z_3 - g_3(z_1, g_2(z_1, \mathbf{u}) + (z_2 - g_2(z_1, \mathbf{u})), \mathbf{u})}_{h_3^{-1}(z_1, z_2, z_3, \mathbf{u})}. \tag{15}$$

$$......,$$

Again, since the composition of polynomials is still a polynomial, the mapping $\mathbf{h}^{-1}$ is also a polynomial.

**Lemma A.3** *The mapping from* $\mathbf{n}$ *to* $\mathbf{x}$, *e.g.,* $\mathbf{f} \circ \mathbf{h}$, *is invertible, and the Jacobi determinant* $|\det \mathbf{J}_{\mathbf{f} \circ \mathbf{h}}| = |\det \mathbf{J}_{\mathbf{f}}||\det \mathbf{J}_{\mathbf{h}}| = |\det \mathbf{J}_{\mathbf{f}}|$, *and thus* $|\det \mathbf{J}_{(\mathbf{f} \circ \mathbf{h})^{-1}}| = |\det \mathbf{J}_{\mathbf{f} \circ \mathbf{h}}^{-1}| = |\det \mathbf{J}_{\mathbf{f}}^{-1}|$, *which do not depend on* $\mathbf{u}$.

Proof can be easily shown by the following: Lemma A.2 has shown that the mapping $\mathbf{h}$, from $\mathbf{n}$ to $\mathbf{z}$, is invertible. Together with the assumption that $\mathbf{f}$ is invertible, the mapping from $\mathbf{n}$ to $\mathbf{x}$ is invertible. In addition, due to the additive noise models and DAG constraint as defined in Eq. 14, we can obtain $|\det \mathbf{J}_{\mathbf{h}}| = 1$.

**Lemma A.4** *Given the assumption (iv) in Theorem 3.1, the partial derivative of* $h_i(n_1, ..., n_i, \mathbf{u})$ *in Eq. 14 with respect to* $n_{i'}$, *where* $i' < i$, *equals 0 when* $\mathbf{u} = \mathbf{0}$, *i.e.,* $\frac{\partial h_i(n_1, ..., n_i, \mathbf{u}=\mathbf{0})}{\partial n_{i'}} = 0$.

Since the partial derivative of the polynomial $h_i(n_1, ..., n_i, \mathbf{u})$ is still a polynomial whose coefficients are scaled by $\boldsymbol{\lambda}_i(\mathbf{u})$, as defined in Eq. 14, and by using the assumption (iv), we can obtain the result.

The proof of Theorem 3.1 is done in three steps. Step I is to show that the identifiability result in (Sorrenson et al., 2020) holds in our setting, i.e., the latent noise variables $\mathbf{n}$ can be identified up

to component-wise scaling and permutation, $\mathbf{n} = \mathbf{P}\hat{\mathbf{n}} + \mathbf{c}$. Using this result, Step II shows that $\mathbf{z}$ can be identified up to polynomial transformation, i.e., $\mathbf{z} = Poly(\hat{\mathbf{z}}) + \mathbf{c}$. Step III shows that the polynomial transformation in Step II can be reduced to permutation and scaling, $\mathbf{z} = \mathbf{P}\hat{\mathbf{z}} + \mathbf{c}$, by using Lemma A.4.

**Step I:** Suppose we have two sets of parameters $\boldsymbol{\theta} = (\mathbf{f}, \mathbf{T}, \boldsymbol{\lambda}, \boldsymbol{\eta})$ and $\hat{\boldsymbol{\theta}} = (\hat{\mathbf{f}}, \hat{\mathbf{T}}, \hat{\boldsymbol{\lambda}}, \hat{\boldsymbol{\eta}})$ corresponding to the same conditional probabilities, i.e., $p_{(\mathbf{f}, \mathbf{T}, \boldsymbol{\lambda}, \boldsymbol{\eta})}(\mathbf{x}|\mathbf{u}) = p_{(\hat{\mathbf{f}}, \hat{\mathbf{T}}, \hat{\boldsymbol{\lambda}}, \hat{\boldsymbol{\eta}})}(\mathbf{x}|\mathbf{u})$ for all pairs $(\mathbf{x}, \mathbf{u})$, where $\mathbf{T}$ denote the sufficient statistic of latent noise variables $\mathbf{n}$. Due to the assumption (i), the assumption (ii), and the fact that $\mathbf{h}$ is invertible (e.g., Lemma A.2), by expanding the conditional probabilities (More details can be found in Step I for proof of Theorem 1 in (Khemakhem et al., 2020)), we have:

$$\log|\det \mathbf{J}_{(\mathbf{f} \circ \mathbf{h})^{-1}}(\mathbf{x})| + \log p_{(\mathbf{T}, \boldsymbol{\eta})}(\mathbf{n}|\mathbf{u}) = \log|\det \mathbf{J}_{(\hat{\mathbf{f}} \circ \hat{\mathbf{h}})^{-1}}(\mathbf{x})| + \log p_{(\hat{\mathbf{T}}, \hat{\boldsymbol{\eta}})}(\hat{\mathbf{n}}|\mathbf{u}), \quad (16)$$

Using the exponential family as defined in Eq. 1, we have:

$$\log|\det \mathbf{J}_{(\mathbf{f} \circ \mathbf{h})^{-1}}(\mathbf{x})| + \mathbf{T}^T\big((\mathbf{f} \circ \mathbf{h})^{-1}(\mathbf{x})\big)\boldsymbol{\eta}(\mathbf{u}) - \log \prod_i Z_i(\mathbf{u}) = \quad (17)$$

$$\log|\det \mathbf{J}_{(\hat{\mathbf{f}} \circ \hat{\mathbf{h}})^{-1}}(\mathbf{x})| + \hat{\mathbf{T}}^T\big((\hat{\mathbf{f}} \circ \hat{\mathbf{h}})^{-1}(\mathbf{x})\big)\hat{\boldsymbol{\eta}}(\mathbf{u}) - \log \prod_i \hat{Z}_i(\mathbf{u}), \quad (18)$$

By using Lemma A.3, Eqs. 17-18 can be reduced to:

$$\log|\det \mathbf{J}_{\mathbf{f}^{-1}}(\mathbf{x})| + \mathbf{T}^T\big((\mathbf{f} \circ \mathbf{h})^{-1}(\mathbf{x})\big)\boldsymbol{\eta}(\mathbf{u}) - \log \prod_i Z_i(\mathbf{u}) =$$

$$\log|\det \mathbf{J}_{\hat{\mathbf{f}}^{-1}}(\mathbf{x})| + \hat{\mathbf{T}}^T\big((\hat{\mathbf{f}} \circ \hat{\mathbf{h}})^{-1}(\mathbf{x})\big)\hat{\boldsymbol{\eta}}(\mathbf{u}) - \log \prod_i \hat{Z}_i(\mathbf{u}). \quad (19)$$

Then by expanding the above at points $\mathbf{u}_l$ and $\mathbf{u}_0$, then using Eq. 19 at point $\mathbf{u}_l$ subtract Eq. 19 at point $\mathbf{u}_0$, we find:

$$\langle \mathbf{T}(\mathbf{n}), \bar{\boldsymbol{\eta}}(\mathbf{u}) \rangle + \sum_i \log \frac{Z_i(\mathbf{u}_0)}{Z_i(\mathbf{u}_l)} = \langle \hat{\mathbf{T}}(\hat{\mathbf{n}}), \bar{\hat{\boldsymbol{\eta}}}(\mathbf{u}) \rangle + \sum_i \log \frac{\hat{Z}_i(\mathbf{u}_0)}{\hat{Z}_i(\mathbf{u}_l)}. \quad (20)$$

Here $\bar{\boldsymbol{\eta}}(\mathbf{u}_l) = \boldsymbol{\eta}(\mathbf{u}_l) - \boldsymbol{\eta}(\mathbf{u}_0)$. By assumption (iii), and combining the $2\ell$ expressions into a single matrix equation, we can write this in terms of $\mathbf{L}$ from assumption (iii),

$$\mathbf{L}^T \mathbf{T}(\mathbf{n}) = \hat{\mathbf{L}}^T \hat{\mathbf{T}}(\hat{\mathbf{n}}) + \mathbf{b}. \quad (21)$$

Since $\mathbf{L}^T$ is invertible, we can multiply this expression by its inverse from the left to get:

$$\mathbf{T}\big((\mathbf{f} \circ \mathbf{h})^{-1}(\mathbf{x})\big) = \mathbf{A}\hat{\mathbf{T}}\big((\hat{\mathbf{f}} \circ \hat{\mathbf{h}})^{-1}(\mathbf{x})\big) + \mathbf{c}, \quad (22)$$

Where $\mathbf{A} = (\mathbf{L}^T)^{-1}\hat{\mathbf{L}}^T$. According to lemma 3 in (Khemakhem et al., 2020) that there exist $k$ distinct values $n_i^1$ to $n_i^k$ such that the derivative $T'(n_i^1), ..., T'(n_i^k)$ are linearly independent, and the fact that each component of $T_{i,j}$ is univariate, we can show that $\mathbf{A}$ is invertible.

Since we assume the noise to be two-parameter exponential family members, Eq. 22 can be re-expressed as:

$$\begin{pmatrix} \mathbf{T}_1(\mathbf{n}) \\ \mathbf{T}_2(\mathbf{n}) \end{pmatrix} = \mathbf{A} \begin{pmatrix} \hat{\mathbf{T}}_1(\hat{\mathbf{n}}) \\ \hat{\mathbf{T}}_2(\hat{\mathbf{n}}) \end{pmatrix} + \mathbf{c}, \quad (23)$$

Then, we re-express $\mathbf{T}_2$ in term of $\mathbf{T}_1$, e.g., $T_2(n_i) = t(T_1(n_i))$ where $t$ is a nonlinear mapping. As a result, we have from Eq. 23 that: (a) $T_1(n_i)$ can be linear combination of $\hat{\mathbf{T}}_1(\hat{\mathbf{n}})$ and $\hat{\mathbf{T}}_2(\hat{\mathbf{n}})$, and (b) $t(T_1(n_i))$ can also be linear combination of $\hat{\mathbf{T}}_1(\hat{\mathbf{n}})$ and $\hat{\mathbf{T}}_2(\hat{\mathbf{n}})$. This implies the contradiction that both $T_1(n_i)$ and its nonlinear transformation $t(T_1(n_i))$ can be expressed by linear combination of $\hat{\mathbf{T}}_1(\hat{\mathbf{n}})$ and $\hat{\mathbf{T}}_2(\hat{\mathbf{n}})$. This contradiction leads to that $\mathbf{A}$ can be reduced to permutation matrix $\mathbf{P}$ (See APPENDIX C in (Sorrenson et al., 2020) for more details):

$$\mathbf{n} = \mathbf{P}\hat{\mathbf{n}} + \mathbf{c}, \quad (24)$$

where $\mathbf{P}$ denote the permutation matrix with scaling, $\mathbf{c}$ denote a constant vector. Note that this result holds for not only Gaussian, but also inverse Gaussian, Beta, Gamma, and Inverse Gamma (See Table 1 in (Sorrenson et al., 2020)).

**Step II:** By Lemma A.2, we can denote $\mathbf{z}$ and $\hat{\mathbf{z}}$ by:

$$\mathbf{z} = \mathbf{h}(\mathbf{n}), \tag{25}$$

$$\hat{\mathbf{z}} = \hat{\mathbf{h}}(\hat{\mathbf{n}}), \tag{26}$$

where $\mathbf{h}$ is defined in A.2. Replacing $\mathbf{n}$ and $\hat{\mathbf{n}}$ in Eq. 24 by Eq. 25 and Eq. 26, respectively, we have:

$$\mathbf{h}^{-1}(\mathbf{z}) = \mathbf{P}\hat{\mathbf{h}}^{-1}(\hat{\mathbf{z}}) + \mathbf{c}, \tag{27}$$

where $\mathbf{h}$ (as well as $\hat{\mathbf{h}}$) are invertible supported by Lemma A.2. We can rewrite Eq. 27 as:

$$\mathbf{z} = \mathbf{h}(\mathbf{P}\hat{\mathbf{h}}^{-1}(\hat{\mathbf{z}}) + \mathbf{c}). \tag{28}$$

Again, by the fact that the composition of polynomials is still a polynomial, we can show:

$$\mathbf{z} = Poly(\hat{\mathbf{z}}) + \mathbf{c}'. \tag{29}$$

**Step III** Next, Replacing $\mathbf{z}$ and $\hat{\mathbf{z}}$ in Eq. 29 by Eqs. 24, 25, and 26:

$$\mathbf{h}(\mathbf{P}\hat{\mathbf{n}} + \mathbf{c}) = Poly(\hat{\mathbf{h}}(\hat{\mathbf{n}})) + \mathbf{c}' \tag{30}$$

By differentiating Eq. 30 with respect to $\hat{\mathbf{n}}$

$$\mathbf{J}_{\mathbf{h}}\mathbf{P} = \mathbf{J}_{Poly}\mathbf{J}_{\hat{\mathbf{h}}}. \tag{31}$$

Without loss of generality, let us consider the correct causal order $z_1 \succ z_2 \succ ..., \succ z_\ell$ so that $\mathbf{J}_{\mathbf{h}}$ and $\mathbf{J}_{\hat{\mathbf{h}}}$ are lower triangular matrices whose the diagonal are 1, and $\mathbf{P}$ is a diagonal matrix with elements $s_{1,1}, s_{2,2}, s_{3,3}, ....$

**Elements above the diagonal of matrix $\mathbf{J}_{Poly}$** Since $\mathbf{J}_{\hat{\mathbf{h}}}$ are lower triangular matrices, and $\mathbf{P}$ is a diagonal matrix, $\mathbf{J}_{Poly}$ must be a lower triangular matrix.

Then by expanding the left side of Eq. 31, we have:

$$\mathbf{J}_{\mathbf{h}}\mathbf{P} = \begin{pmatrix} s_{1,1} & 0 & 0 & ... \\ s_{1,1}\frac{\partial h_2(n_1,n_2,\mathbf{u})}{\partial n_1} & s_{2,2} & 0 & ... \\ s_{1,1}\frac{\partial h_3(n_1,n_2,n_3,\mathbf{u})}{\partial n_1} & s_{2,2}\frac{\partial h_3(n_1,n_2,n_3,\mathbf{u})}{\partial n_2} & s_{3,3} & ... \\ . & . & . & ... \end{pmatrix}, \tag{32}$$

by expanding the right side of Eq. 31, we have:

$$\mathbf{J}_{Poly}\mathbf{J}_{\hat{\mathbf{h}}} = \begin{pmatrix} J_{Poly_{1,1}} & 0 & 0 & ... \\ J_{Poly_{2,1}} + J_{Poly_{2,2}}\frac{\partial \hat{h}_2(n_1,n_2,\mathbf{u})}{\partial n_1} & J_{Poly_{2,2}} & 0 & ... \\ J_{Poly_{3,1}} + \sum_{i=2}^{3} J_{Poly_{3,i}}\frac{\partial \hat{h}_i(n_1,...,n_i,\mathbf{u})}{\partial n_1} & J_{Poly_{3,2}} + J_{Poly_{3,3}}\frac{\partial \hat{h}_3(n_1,...,n_3,\mathbf{u})}{\partial n_2} & J_{Poly_{3,3}} & ... \\ . & . & . & ... \end{pmatrix}. \tag{33}$$

**The diagonal of matrix $\mathbf{J}_{Poly}$** By comparison between Eq. 32 and Eq. 33, we have $J_{Poly_{i,i}} = s_{i,i}$

**Elements below the diagonal of matrix $\mathbf{J}_{Poly}$** By comparison between Eq. 32 and Eq. 33, and Lemma A.4, for all $i > j$ we have $J_{Poly_{i,j}} = 0$.

As a result, the matrix $\mathbf{J}_{Poly}$ in Eq. 31 equals to the permutation matrix $\mathbf{P}$, which implies that the polynomial transformation Eq. 29 reduces to a permutation transformation,

$$\mathbf{z} = \mathbf{P}\hat{\mathbf{z}} + \mathbf{c}'. \tag{34}$$

A.3    THE PROOF OF COROLLARY 3.2

To prove the corollary, we demonstrate that it is always possible to construct an alternative solution, which is different from the true $\mathbf{z}$, but capable of generating the same observations $\mathbf{x}$, if there is an unchanged coefficient across $\mathbf{u}$. Again, without loss of the generality, suppose that the correct causal order is $z_1 \succ z_2 \succ ... \succ z_\ell$. Suppose that for $z_i$, there is an unchanged coefficient $\lambda_{j,i}$, related to the term of polynomial $\lambda_{j,i}\phi$ across $\mathbf{u}$, where $\phi$ denotes polynomial features created by raising the variables related to parent node to an exponent. Note that since we assume the correct causal order, the term $\phi$ only includes $z_j$ where $j < i$. Then, we can always construct new latent variables $\mathbf{z}'$ as: for all $k \neq i$, $z'_k = z_k$, and $z'_i = z_i - \lambda_{j,i}\phi$. Given this, we can construct a polynomial mapping $\mathbf{M}$, so that

$$\mathbf{M}(\mathbf{z}') = \mathbf{z}, \tag{35}$$

where

$$\mathbf{M}(\mathbf{z}') = \begin{pmatrix} z'_1 \\ z'_2 \\ .. \\ z'_i + \lambda_{j,i}\phi \\ z'_{i+1} = z_{i+1} \\ .. \end{pmatrix}. \tag{36}$$

where for all $k \neq i$, $z'_k = z_k$ in the right. It is clear the Jacobi determinant of the mapping $\mathbf{M}$ always equals 1, thus the mapping $\mathbf{M}$ is invertible. In addition, all the coefficients of the polynomial mapping $\mathbf{M}$ are constant and thus do not depend on $\mathbf{u}$. As a result, we can construct a mapping $\mathbf{f} \circ \mathbf{M}$ as the mapping from $\mathbf{z}'$ to $\mathbf{x}$, which is invertible and do not depend on $\mathbf{u}$, and can create the same data $\mathbf{x}$ generated by $\mathbf{f}(\mathbf{z})$. Therefore, the alternative solution $\mathbf{z}'$ can lead to a non-identifiability result.

A.4    THE PROOF OF COROLLARY 3.3

Since the proof process in Steps I and II in A.2 do not depend on the assumption of change of causal influence, the results in both Eq. 32 and Eq. 33 hold. Then consider the following two cases.

- For the case where $z_i$ is a root node or all coefficients on all paths from parent nodes to $z_i$ change across $\mathbf{u}$, by using Lemma A.4, i.e., $\frac{\partial h_i(n_1,...,n_i,\mathbf{u=0})}{\partial n_{i'}} = 0$ and $\frac{\partial \hat{h}_i(n_1,...,n_i,\mathbf{u=0})}{\partial n_{i'}} = 0$ for all $i' < i$, and by comparison between Eq. 32 and Eq. 33, we have: for all $i > j$ we have $J_{Poly_{i,j}} = 0$, which implies that we can obtain that $z_i = A_{i,i}\hat{z}_i + c'_i$.

- If there exists an unchanged coefficient in all paths from parent nodes to $z_i$ across $\mathbf{u}$, then by the proof of corollary 3.2, i.e., $z'_i$ can be constructed as a new possible solution to replace $z_i$ by removing the unchanged coefficient $\lambda_{j,i}$, resulting in the non-identifiability result. This is also can be proved in another way, e.g., a comparison between Eq. 32 and Eq. 33. Suppose that the coefficient is $\lambda_{j,i}$ related to the parent node with the index $k$. Given that, we have: $\frac{\partial \hat{h}_i(n_1,...,n_{i-1},\mathbf{u})}{\partial n_k}$ include a constant term $\lambda_{j,i}$. Again, by using the Lemma A.4, and by comparison between Eq. 32 and Eq. 33, we can only arrive that $s_{k,k}\lambda_{j,i} = J_{Poly_{i,k}}$. As a result, $z_i$ will be expressed as a combination of $\hat{z}_k$ and $\hat{z}_i$.

## A.5 SYNTHETIC DATA

**Data** For experimental results on synthetic data, the number of segments is 30, and for each segment, the sample size is 1000, while the number (*e.g.*, dimension) of latent causal or noise variables is 2,3,4,5 respectively. Specifically, for latent linear causal models, we consider the following structural causal model:

$$n_i :\sim \begin{cases} \mathcal{B}(\alpha, \beta), & \text{if } \mathbf{n} \sim \text{Beta} \\ \mathcal{G}(\alpha, \beta),, & \text{if } \mathbf{n} \sim \text{Gamma} \end{cases} \tag{37}$$

$$z_1 := n_1 \tag{38}$$
$$z_2 := \lambda_{1,2}(\mathbf{u})z_1 + n_2 \tag{39}$$
$$z_3 := \lambda_{2,3}(\mathbf{u})z_2 + n_3 \tag{40}$$
$$z_4 := \lambda_{3,4}(\mathbf{u})z_3 + n_4 \tag{41}$$
$$z_5 := \lambda_{3,5}(\mathbf{u})z_3 + n_5, \tag{42}$$
$$\tag{43}$$

where both $\alpha$ and $\beta$, for both Beta and Gamma distributions, are sampled from a uniform distribution $[0.1, 2.0]$, and $\lambda_{i,j}(\mathbf{u})$ are sampled from a uniform distribution $[-1.0, -0.5] \cup [0.5, 1.0]$. For latent polynomial causal models with Gaussian noise, we consider the following structural causal model:

$$n_i :\sim \mathcal{N}(\alpha, \beta), \tag{44}$$
$$z_1 := n_1 \tag{45}$$
$$z_2 := \lambda_{1,2}(\mathbf{u})z_1^2 + n_2 \tag{46}$$
$$z_3 := \lambda_{2,3}(\mathbf{u})z_2 + n_3 \tag{47}$$
$$z_4 := \lambda_{3,4}(\mathbf{u})z_2 z_3 + n_4 \tag{48}$$
$$z_5 := \lambda_{3,5}(\mathbf{u})z_3^2 + n_5. \tag{49}$$
$$\tag{50}$$

where both $\alpha$ and $\beta$ for Gaussian noise are sampled from uniform distributions $[-2.0, 2.0]$ and $[0.1, 2.0]$, respectively. $\lambda_{i,j}(\mathbf{u})$ are sampled from a uniform distribution $[-1.0, -0.5] \cup [0.5, 1.0]$.

## A.6 IMPLEMENTATION FRAMEWORK

Figure 8 depicts the proposed method to learn polynomial causal representations with *non-Gaussian noise*. Figure 9 depicts the proposed method to learn polynomial causal representations with *Gaussian noise*. For non-Gaussian noise, since there is generally no analytic form for the joint distribution of latent causal variables, we here assume that $p(\mathbf{z}|\mathbf{u}) = p(\boldsymbol{\lambda}(\mathbf{u}))p(\mathbf{n}|\mathbf{u})$ as defined in Eq. 6. Note that this assumption may be not true in general, since it destroys independent causal mechanisms generating effects in causal systems. For experiments on the synthetic data and fMRI data, the encoder, decoder, MLP for $\boldsymbol{\lambda}$, and MLP for prior are implemented by using 3-layer fully connected networks and Leaky-ReLU activation functions. For optimization, we use Adam optimizer with learning rate 0.001. For experiments on the image data, we also use 3-layer fully connected network and Leaky-ReLU activation functions for $\boldsymbol{\lambda}$ and the prior model. The encoder and decoder can be found in Table 1 and Table 2, respectively.

| Input |
|---|
| Leaky-ReLU(Conv2d(3, 32, 4, stride=2, padding=1)) |
| Leaky-ReLU(Conv2d(32, 32, 4, stride=2, padding=1)) |
| Leaky-ReLU(Conv2d(32, 32, 4, stride=2, padding=1)) |
| Leaky-ReLU(Conv2d(32, 32, 4, stride=2, padding=1)) |
| Leaky-ReLU(Linear(32×32×4 + size($\mathbf{u}$), 30)) |
| Leaky-ReLU(Linear(30, 30)) |
| Linear(30, 3*2) |

Table 1: Encoder for the image data.

| Latent Variables |
|---|
| Leaky-ReLU(Linear(3, 30)) |
| Leaky-ReLU(Linear(30, 30)) |
| Leaky-ReLU(Linear(30, 32 × 32 ×4)) |
| Leaky-ReLU(ConvTranspose2d(32, 32, 4, stride=2, padding=1)) |
| Leaky-ReLU(ConvTranspose2d(32, 32, 4, stride=2, padding=1)) |
| Leaky-ReLU(ConvTranspose2d(32, 32, 4, stride=2, padding=1)) |
| ConvTranspose2d(32, 3, 4, stride=2, padding=1) |

Table 2: Decoder for the image data.

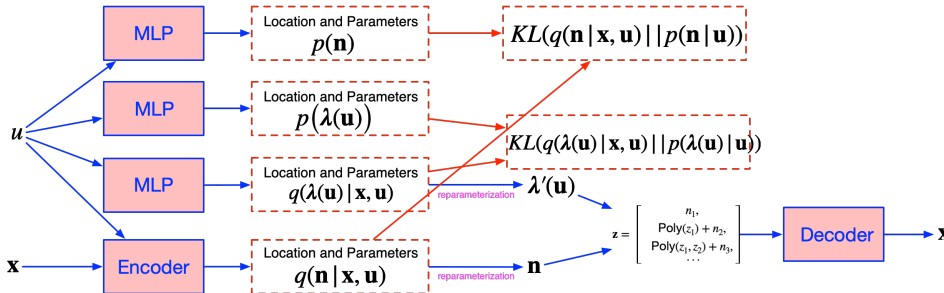

Figure 8: Implementation Framework to learn polynomial causal representations with non-Gaussian noise.

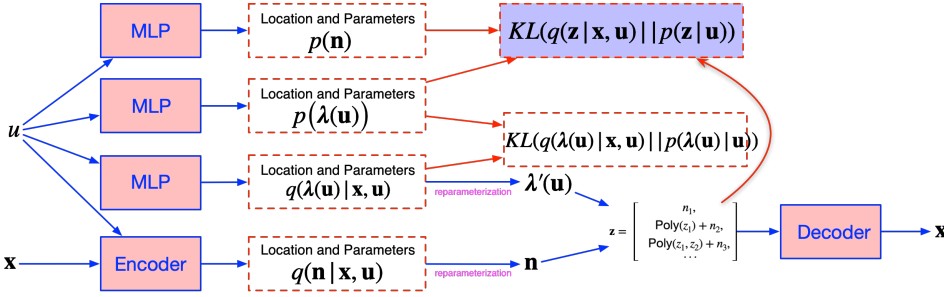

Figure 9: Implementation Framework to learn polynomial causal representations with Gaussian noise, for which an analytic form for the prior and posterior of latent causal variables can be provided.

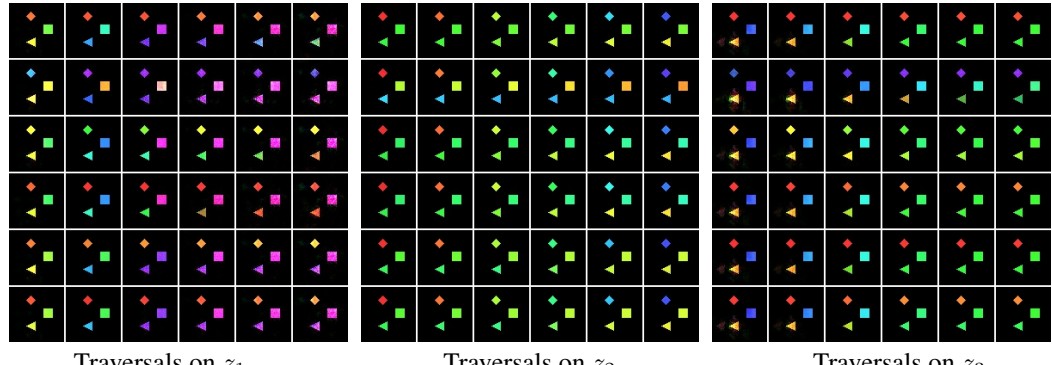

Figure 10: Traversal results obtained by $\beta$-VAE on the image data. The vertical axis denotes different samples, The horizontal axis denotes enforcing different values on the learned causal representation. The ground truth of the latent causal graph is that the 'diamond' (e.g., $z_1$) causes the 'triangle' (e.g., $z_2$), and the 'triangle' causes the 'square' (e.g., $z_3$). We can see that the change of each learned variable results in the change of color of all objects.

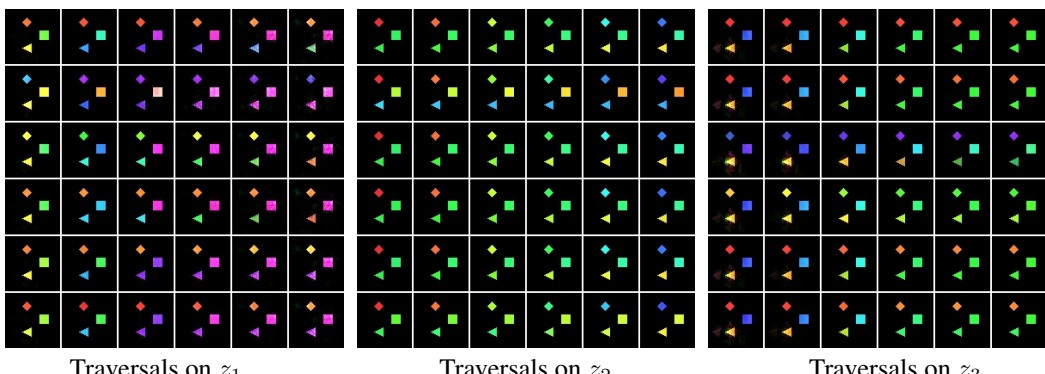

Figure 11: Traversal results obtained by VAE on the image data. The vertical axis denotes different samples, The horizontal axis denotes enforcing different values on the learned causal representation. The ground truth of the latent causal graph is that the 'diamond' (e.g., $z_1$) causes the 'triangle' (e.g., $z_2$), and the 'triangle' causes the 'square' (e.g., $z_3$). We can see that the change of each learned variable results in the change of color of all objects.

## A.7 TRAVERSALS ON THE LEARNED VARIABLES BY IVAE, $\beta$-VAE, AND VAE

Since there is no theoretical support for both $\beta$-VAE and VAE, these two methods can not disentangle latent causal variables. This can be demonstrated by Figure 10 and Figure 11, which shows that traversal on each learned variable leads to the change of colors of all objects. It is interesting to note that iVAE has the theoretical guarantee of learning latent noise variables. And since we assume additive noise models, e.g., $z_1 = n_1$, iVAE is able to identify $z_1$. This can be verified from the result obtained by iVAE shown in Figure 5, which shows that the MPC between recovered $z_1$ and the true one is 0.963. However, iVAE can not identify $z_2$ and $z_3$, since these are causal relations among the latent causal variables. We can see from Figure 12, the changes of $z_2$ and $z_3$ lead to the change of colors of all objects.

## A.8 FURTHER DISCUSSION ON THE PARTIAL IDENTIFIABILITY IN COROLLARY 3.3.

While demanding a change in all coefficients may pose challenges in practical applications, Corollary 3.3 introduces partial identifiability results. The entire latent space can theoretically be partitioned into two distinct subspaces: an invariant latent subspace and a variant latent subspace. This partitioning holds potential value for applications emphasizing the learning of invariant latent variables to adapt to changing environments, such as domain adaptation (or generalization), as discussed

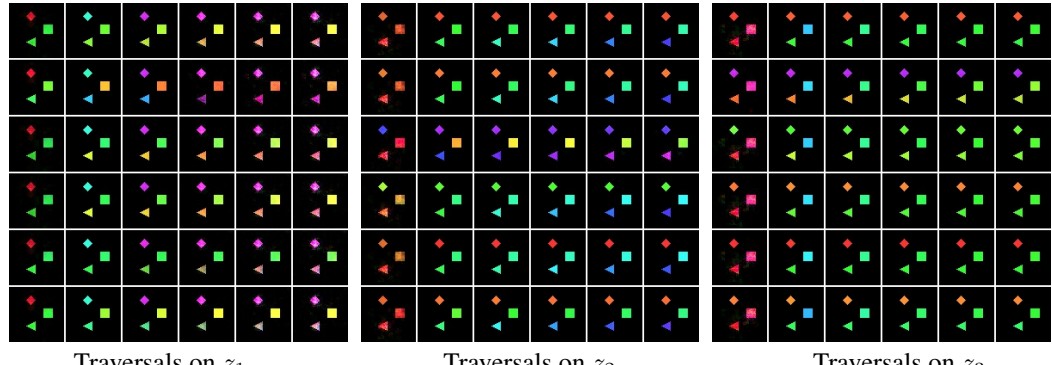

Traversals on $z_1$ · · · Traversals on $z_2$ · · · Traversals on $z_3$

Figure 12: Traversal results obtained by iVAE on the image data. The vertical axis denotes different samples, The horizontal axis denotes enforcing different values on the learned causal representation. The ground truth of the latent causal graph is that the 'diamond' (e.g., $z_1$) causes the 'triangle' (e.g., $z_2$), and the 'triangle' causes the 'square' (e.g., $z_3$). We can see that the change of each learned variable results in the change of color of all objects.

in the main paper. However, the impact of partial identifiability results on the latent causal graph structure remains unclear.

We posit that if there are no interactions (edges) between the two latent subspaces in the ground truth graph structure, the latent causal structure in the latent variant space can be recovered. This recovery is possible since the values of these variant latent variables can be restored up to component-wise permutation and scaling. In contrast, when no interactions exist between the two latent subspaces in the ground truth graph structure, recovering (part of) the latent causal structure becomes highly challenging. We believe that the unidentifiable variables in the invariant latent subspace may influence the latent causal structure in the variant latent subspace.

We hope this discussion can inspire further research to explore this intriguing problem in the future.

## A.9 FURTHER DISCUSSION ON MODEL ASSUMPTIONS IN LATENT SPACE.

In our approach, we selected polynomial models for their approximation capabilities and straightforward expressions, streamlining analysis and facilitating the formulation of sufficient changes. While advantageous, this choice is not without recognized limitations, notably the challenges introduced by high-order terms in polynomials during optimization. Overall, we think that model assumptions can be extended to a broader scope than polynomials, including non-parametric models. This extension is contingent on deeming changes in causal influences as sufficient. The crucial question in moving towards more general model assumptions revolves around defining what constitutes sufficient changes in causal influences.

## A.10 UNDERSTANDING ASSUMPTIONS IN THEOREM

Assumptions (i)-(iii) are motivated by the nonlinear ICA literature (Khemakhem et al., 2020), which is to provide a guarantee that we can recover latent noise variables $\mathbf{n}$ up to a permutation and scaling transformation. The main Assumption (iii) essentially requires sufficient changes in latent noise variables to facilitate their identification. Assumption (iv) is derived from the work by Liu et al. (2022) and is introduced to avoid a specific case: $\lambda_{i,j}(\mathbf{u}) = \lambda'_{i,j}(\mathbf{u}) + b$, where $b$ is a non-zero constant. To illustrate, if $z_2 = (\lambda'_{1,2}(\mathbf{u}) + b)z_1 + n_2$, the term $bz_1$ remains unchanged across all $\mathbf{u}$, resulting in non-identifiability according to Corollary 3.3. While assumption (iv) is sufficient for handling this specific case, it may not be necessary. We anticipate the proposal of a sufficient and necessary condition in the future to address the mentioned special case.

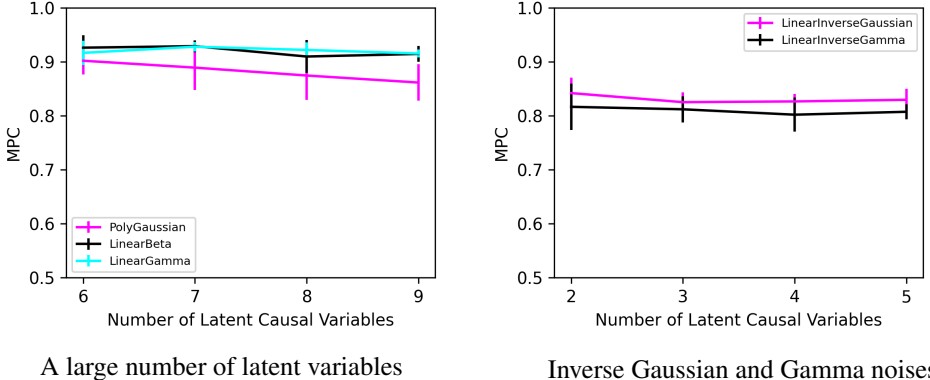

A large number of latent variables     Inverse Gaussian and Gamma noises

Figure 13: Performances of the proposed method on a large number of latent variables and latent linear models with inverse Gaussian and inverse Gamma noises.

## A.11 THE UNKNOWN NUMBER OF LATENT CAUSAL/NOISE VARIABLES

It is worth noting that most existing works require knowledge of the number of latent variables. However, this is not a theoretical hurdle in our work. It is due to a crucial step in our results leveraging the identifiability findings from Sorrenson et al. (2020) to idenitify latent noise variables. The key insight in Sorrenson et al. (2020) demonstrates that the dimension of the generating latent space can be recovered if latent noises are sampled from the two-parameter exponential family members. This assumption is consistent with our assumption on latent noise, as defined in Eq. 1. In other words, if the estimated latent space has a higher dimension than the generating latent space, some estimated latent variables may not be related to any generating latent variables and therefore encode only noise. More details can be found in Sorrenson et al. (2020).

## A.12 MORE RESULTS AND DISCUSSION

In this section, we present additional experimental results conducted on synthetic data to demonstrate the effectiveness of the proposed method across scenarios involving a large number of latent variables as well as latent variables characterized by inverse Gaussian and inverse Gamma distributions. Both scenarios present significant challenges in optimization. The performance on these cases is depicted in Figure 13. In the left subfigure of Figure 13, for ployGaussian, the presence of numerical instability and exponential growth in polynomial model terms complicates the recovery of latent polynomial causal models, especially with an increasing number of latent variables. Additionally, the reparameterization trick for inverse Gaussian and inverse Gamma distributions poses challenges in recovering latent linear models. Both cases warrant further exploration in future research.

