# OpenReview forum: "Identifiable Latent Polynomial Causal Models through the Lens of Change"
_ICLR.cc/2024/Conference — ICLR 2024 poster_

### Official Review · Reviewer_YEeM · 2023-10-26

**Soundness:** 3 good
**Presentation:** 4 excellent
**Contribution:** 3 good
**Rating:** 6
**Confidence:** 3

**Summary:**

This paper focuses on identifying a causal graph in the latent space. The paper extends previous methods as follows: 1) the dependencies between the latent variables can be polynomial (previously linear), 2) the noise of the latent variables can be in the exponencial family (previously Gaussian). The paper presents a theorem for the identifiability of such models, which in general is based on the principle of observing the system in multiple different environments. The number of environments required in the proof is smaller than previous proofs (2l+1 environments vs. the previous quadratic number). The paper also presents proofs for partial (un)identifiability, basically stating that for those parts of the model where the environments behave differently the latent variable can be learned, otherwise the latent variable is unidentifiable.

My overall initial view of this article is cautiously positive. However, I have multiple questions for clarifications.

**Strengths:**

1) The paper is very well written. The background is clearly explained and sufficient. Sketches of proofs are given for the theorems, which nicely give intuition. Mathematical parts (model, theorems, training) are rigorously presented. Overall, the presentation makes a nice trade-off between accessibility and rigorousness.
2) The topic of latent causal modeling is challenging and the paper makes a few novel contributions in this challenging domain.
3) The experiments are reasonably thorough and demonstrate and complement the theoretical developments in an intuitive way.

**Weaknesses:**

1) The paper relies quite a bit on a previous (unpublished) paper (Liu et al., 2022), which affects novelty. Overall, the paper makes a few contributions but none of these seems strikingly novel or signficant. However, the overall significance/novelty seems sufficient for acceptance in my opinion.
2) The empirical experiments are somewhat artificial, and do not demonstrate real-world relevance (often the case in papers in this domain). The results are only shown for those setups where the method worked: LinearGamma, LinearBeta, PolyGamma, and not for the difficult cases: e.g. inverse Gamma etc. These could have been included in the Appendix at least, even if they didn’t work well (which the paper openly mentions).
3) Some assumptions (additivity, knowledge of the number of latent variables) seem quite strong.

**Questions:**

1) Could you comment on how to obtain the number of latent variables in practice? You are assuming it’s known in this paper, right?
2) Condition (iii) in Thm 3.1 has u_{2l+1} although u’s exist only until 2l. Possible typo?
3) Could you include additional intuition for conditions (iii) and (iv) of Thm 3.1?
4) Proof sketch of Thm 3.1 says  “arises from the fact that additive noise models are identifiable”. If I remember correctly, additive linear Gaussian models are not identifiable? Could you clarify how this is compatible with the sentence?
5) About Corollary 3.2: There is a condition: “if there is an unchanged coefficient in Eq 4 across environments, z is unidentifiable.”. What prevented such a condition from happening in Theorem 3.1?
6) Second line of Equation (6): Where is z hidden on the RHS of the equation?
7) Prior on lambda: you encure acyclicity by assuming lambda is lower-dimensional. Does this mean that you have to assume the causal order?
8) MPC metric: correlation between what exactly?
9) The last sentence of Discussion: “even with identifiability guarantees, it is still challenging to learn causal graphs in latent space”. It would have been interesting to explore how the number of samples affects this. Do I understand it correctly that the theory guarantees that in the limit of large number of samples the correct graph should be recovered?

---

> ### Author Response · Authors · 2023-11-19
> **Response**
>
> **Q1: the difficult cases: e.g. inverse Gamma etc, could have been included ..**
>
> **R1**: Thank you for your suggestion. Given the constraints of the rebuttal time frame, we must prioritize addressing more complex reviews. We will add these experiments in the final verision. Your understanding is greatly appreciated.
>
> **Q2: How to obtain the number of latent variables?**
>
> **R2**: That is an excellent question. It is worth noting that some existing works may require knowledge of the number of latent variables. However, this is not a theoretical hurdle in our work. It is essential to highlight that a crucial step in our results involves leveraging the identifiability findings from [1] to identify latent noise variables. The key insight in [1] demonstrates that the dimension of the generating latent space can be recovered if latent noises are sampled from the two-parameter exponential family members (consistent with our assumption on latent noise). This is, if the estimated latent space has a higher dimension than the generating latent space, some estimated latent variables are not be related to any generating latent variables and therefore must encode only noise. More details please check the work [1]. We do not highlight this problem in our original version, considering limited space and that it is not our contribution. We add a discussion about it in new version, see Section A. 11.
>
> [1] Sorrenson, Peter, Carsten Rother, and Ullrich Köthe. "Disentanglement by nonlinear ica with general incompressible-flow networks (gin)." arXiv preprint arXiv:2001.04872 (2020).
>
> **Q2: Condition (iii) has u{2l+1} although u’s exist only until 2l. Possible typo?**
>
> **R2**: Please note that the indexing of u starts from 0, not 1.
>
> **Q3: Additional intuition for conditions (iii) and (iv)?**
>
> **R3**:  Condition (iii) stems from nonlinear ICA, it is basically saying that the auxiliary variable must have a sufficiently strong and diverse effect on the distributions of the latent noises, to facilitate their identifiability. Condition (iv) is derived from the work by Liu et al. (2022), aiming to constrain the function class of lambda to prevent a specific case, e.g., lambda(u) = lambda(u) + b, where b is a constant. In this scenario, the invariant part, e.g., b * Pa (parent node), leads to unidentifiability, as discussed in Corollary 3.3. We have added Section A. 10 in the revised verison for understanding our assumptions. Thanks for your suggestion.
>
> **Q4: . additive noise models are identifiable.**
>
> **R4**: It should be "arises from the fact that nonlinear models with additive noise models are identifiable". Thank you for your careful review. In fact, linear Gaussian models in multiple environments are generally identifiable, thanks to the assumption of independent causal mechanisms [1] [2]
>
> [1] Ghassami, AmirEmad, et al. "Multi-domain causal structure learning in linear systems." Advances in neural information processing systems 31 (2018).
>
> [2] Huang, Biwei, et al. "Causal discovery from heterogeneous/nonstationary data." The Journal of Machine Learning Research 21.1 (2020): 3482-3534.
>
> **Q5:  About Corollary 3.2..What prevented such a condition in Theorem 3.1.**
>
> **R5**: ~~Theorem 3.1 claim a complete indeitifbiality result, requiring all all coefficients to change, as modelled by Eq. (4). However, this assumption may appear strong. To assess its necessity, we introduce such a condition in Corollary 3.2 to reinforce the assumption, highlighting its indispensability for identifiability in the absence of any supplementary conditions. Moreover, as the primary motivation in this work is to capture the changes in causal influences, it is crucial to note that, in practice, these changes could be arbitrary. This implies the possibility that some causal influences remain unchanged.~~
>
> **R5**: Assumption (iv) imposes a constraint on the function class, ensuring that $\lambda_{i,j}(\mathbf{u})$ is not non-zero constant. Corollaries 3.2 and 3.3, "Under the condition that the assumption (i)-**(iv)**" should be "Under the condition that the assumption (i)-**(iii)**". Thank you for your thorough review.
>
> **Q6: .Where is z hidden on the RHS..**
>
> **R6**: Again, that is an excellent question. As non-Gaussian noise lacks an analytically tractable solution in general, we model the prior distribution $p(z|u)$ using $p(\lambda, n|u)$, this is because z can be entirely represented by $\lambda$, $n$, and $u$, as further elaborated in Eq. (14). More deeply, from the perspective of independent causal mechanisms, the latent noise variables $n$ need not to be independent of the coefficient $\lambda$, given $u$, since they both correspond to the mechanism generating the effect on latent causal variables. However, considering the challenges in modeling the complicated dependence between $n$ and $\lambda$, and the challenges in optimization, we employ an approximate implementation, as shown in the second line of Eq. (6), which assumes their independence.

---

> > ### Author Response · Authors · 2023-11-19
> > **Response**
> >
> > **Q7: Prior on lambda: you encure acyclicity by assuming lambda is lower-dimensional. Does this mean that you have to assume the causal order?**
> >
> > **R7**: Indeed, we have already assumed the causal order, to avoid trouble DAG constraint, e.g., proposed by Zheng et al.
> > (2018). Essentaily, this is due to the specific permutation indeterminacy in latent space. To illustrate this, let us examine a scenario involving only two latent causal variables, namely size (z1) and color (z2) of an object, with z1 causing z2. Upon obtaining two recovered latent causal variables, z'1 and z'2, permutation indeterminacy arises, allowing z'1 to correspond to either the size (z1) or the color (z2). This flexibility enables us to pre-define a causal order in the inference model, such as z'1 causing z'2, **without specifying semantic information (size or color) for the nodes**. Consequently, the inference model establishes a causal relationship between z'1 and z'2, prompting them to learn the genuine latent semantic information. In other words, the inference model compels z'1 (or z'2) to acquire size (or color) information, effectively sidestepping DAG constraints. For a more comprehensive and detailed explanation, please refer to Section 3.4 of the work by Liu et al. (2022), where the authors provide a thorough clarification of these concepts.
> >
> > **Q8: MPC metric: correlation between what exactly**
> >
> > **R8**: MPC involves computing the correlation between the true $\mathbf{z}$ and the recovered counterpart $\mathbf{\hat z}$. For a $\mathbf{z}$ with a dimension of 5, the computation process begins by generating a 5 $\times$ 5 matrix, where each element represents the Pearson correlation coefficient between $z_i$ and ${\hat z}_j$. Subsequently, a linear sum assignment problem is solved to match each recovered latent variable with the true latent variable that exhibits the highest correlation. This step effectively reverses any permutations in the latent space. The outcome is a 5-dimensional vector, which is then averaged.
> >
> > **Q9: The last sentence of Discussion: “even with identifiability guarantees, it is still challenging to**
> >
> > **R9**: That is an excellent question as well. The challenge stems not only from the influence of sample size, but also, or most importantly, from the optimization problem. In our implementation, we observe that there always be an approximation between the recovered latent causal variables and the true ones, indicating that exact recovery of latent variables is seldom achieved. This prompts another question: could even a minor error in the approximation result in diverse graph structures? To delve deeper, how does the presence of a small error impact the learned graph structure? Regrettably, we are currently unable to provide a definitive answer to this question. Nevertheless, we acknowledge it as a crucial issue deserving attention and further exploration.

---

> > > ### Comment · Reviewer_YEeM · 2023-11-21
> > >
> > > Thanks for your replies. I have no further questions for now. I will reconsider my position after discussing with the other reviewers.

---

> > > > ### Author Response · Authors · 2023-11-22
> > > > **Response**
> > > >
> > > > Appreciate the time and effort you dedicated to reviewing our work.

---

### Official Review · Reviewer_22Xe · 2023-10-30

**Soundness:** 3 good
**Presentation:** 3 good
**Contribution:** 3 good
**Rating:** 6
**Confidence:** 4

**Summary:**

UPDATE: The revised manuscript seems to address the concerns I had, though have not had the opportunity to re-review it. I have increased my score accordingly.

This paper approaches the problem of identifying the latent variable causal model from observed data, by leveraging changes in the causal mechanisms across environments. Identification here means finding the graph as well as the causal mechanisms. The causal mechanisms are generalized compared to previous work, allowing polynomial equations and exponential family noise distributions.

**Strengths:**

The latent causal representation learning problem is important, and I believe the extension proposed here is a significant improvement compared to previous work.

The writing is overall of a good quality. I appreciate the "proof sketch" and "insights" paragraphs following the main results.

**Weaknesses:**

The formal presentation needs work: most importantly, core concepts are ambiguous (see below for details). As a result, I am currently not confident about the soundness of the results. I hope these issues can be addressed with relatively little work.

**Questions:**

"Identifiability": It is expected that it is impossible to exactly reconstruct the latent variables from the observed ones, e.g. due to the possibilities of permutation and scaling. So I would expect that your objective is identification up to certain transformations. However, a definition is not given and it remains unclear what you mean precisely by "identifiability" (and by "unidentifiability", rendering the claims of the corollaries unclear).

Changes across environments: These are mentioned frequently in section 3.3, but it difficult to determine from the text what the precise meaning is: for a coefficient, is it (A) that there exist two environments such that the coefficient is different between those, or (B) that for each pair of distinct environments, the coefficient is different between them?

Theorem 3.1 (i), "measure zero (i.e., has at the most countable number of elements)" - I assume you are referring to Lebesgue measure. In dimension 2 or higher, Lebesgue measure zero does not imply that the number of elements is countable. For instance, the set $\\{(x,0)| x \in \mathbb{R} \\}$ has Lebesgue measure zero, but uncountably many elements.

Theorem 3.1 (iii) - What do you mean by "takes up"? Is it that the vectors $\mathbf{u}_0$ etc are possible values that the random variable $\mathbf{u}$ can take? (I expected that footnote 1 would say something about the distinction between random variables and their values, but I'm not sure that's what it says.)

A "DAG constraint" is first mentioned in the proof sketch of the main theorem. It would be better form to introduce this assumption prior to the proof.

Start of section 3, "even if the causal relationships are nonlinear and noise distributions are non-Gaussian" - isn't the linear Gaussian case the least likely to be identifiable?

What is the $\phi$ in the proof of Corollary 3.2?

Section 4 (and elsewhere): To say that $z_1$ comes first in the causal order, write $z_1 \prec z_2 \prec \ldots$.

Equation (6): What does the $=\mathcal{N}(\ldots)$ mean? It contains unexplained notation, and $z_i$ doesn't appear except in indices.

Other comments (e.g. typographical):
Pages 5 and 7: "identifyable" -> "identifiable"
Page 5: "polynomial propriety" -> "the property"
Corollary 3.2: This statement does not appear to follow from the preceding theorem, so the name "corollary" doesn't seem appropriate.
Appendix A.4: "support that" -> "suppose that"
Appendix A.4, "proof of corollary A.3" -> "proof of corollary 3.2"
End of section 4: "APPENDIX" -> "Appendix"

---

> ### Author Response · Authors · 2023-11-19
> **Response**
>
> **Q1: a definition is not given and it remains unclear what you mean precisely by "identifiability" and by "unidentifiability", rendering the claims of the corollaries unclear.**
>
> **R1**: Please note that the definition of identifiability has already been established in our Theorem 3.1 and in Corollary 3.3, as stated "the true latent causal variables $\mathbf{z}$ are related to the estimated latent causal variables ${ \mathbf{\hat z}}$, which are learned by matching the true marginal data distribution $p(\mathbf{x}|\mathbf{u})$, by the following relationship: $\mathbf{z} = \mathbf{P} \mathbf{\hat z} + \mathbf{c},$ where $\mathbf{P}$ denotes the permutation matrix with scaling, $\mathbf{c}$ denotes a constant vector". Naturally, unidentifiability emerges as the complementary concept to identifiability. That is, z can not be recovered up to $\mathbf{z} = \mathbf{P} \mathbf{\hat z} + \mathbf{c}$.
>
> **Q2: Changes across environments, it difficult to determine from the text what the precise meaning is...**
>
> **R2**: Please be aware that we have formulated the changes using the function $\lambda_{i,j}(u)$ for the coefficients as defined in Eq. (4). This implies that the changes are entirely dependent on the function class of $\lambda_{i,j}$, as long as the function satisfies assumption (iv). Additionally, it is crucial to note that we do not make any claims regarding the pair of environments in our results. Specifically, we do not require paired environments, distinguishing our approach from some prior works that do necessitate such pairing. We have also clarified this aspect in our introduction.
>
> **Q3: measure zero (i.e., has at the most countable number of elements)...**
>
> **R3**: Thank you for such carefully check. In our previous version, we included the phrase 'has at most a countable number of elements' merely for the purpose of easier reader understanding. However, we recognize that this constitutes a stronger assumption than measure zero. In our updated version, we have removed this phrase. Essentially, we here aims to emphasize that $\varphi_{\varepsilon}(\mathbf{x}) \neq 0$ almost everywhere. We appreciate your careful review.
>
> **Q4: What do you mean by "takes up"?...**
>
> **R4**: Thank you for seeking clarification. It's important to note that our assertion does not involve multiple random variables. Our emphasis is on the existence of a single random variable, typically represented by a one-hot vector denoted as $\mathbf{u}$, signifying the environment. Each point or value $\mathbf{u}_i$ represents one environment. This assumption, which is commonplace in nonlinear Independent Component Analysis (ICA) literature, essentially states that the random variable $\mathbf{u}$ must have a sufficiently strong and diverse influence on the distributions of the latent noise variables.
>
> **Q5: DAG constraint, notation, and typos**
>
> **R5**: Thank you for such carefully check. We have carefully modified these problems in new version.
>
> ------------
> Dear Reviewer 22xe,
>
> Your thoughtful reconsideration of our work is highly valued, and we sincerely hope that our clarification provides valuable information, addressing your concerns and enhancing the overall understanding of the contributions and merits of this work. Your thorough re-evaluation is crucial to this work, and we appreciate the opportunity to further engage with any aspects that may contribute to a more comprehensive assessment. Thank you for your time and consideration.

---

> > ### Comment · Reviewer_22Xe · 2023-11-20
> >
> > Thank you for your replies. On the point of Q1, that is a rather implicit way of establishing the definition of a core term; I hope an explicit definition will be added. For Q4, could you say how you would rewrite condition (iii) of theorem 3.1?
> >
> > Also please clarify that in corollary 3.2, equations (1)-(4) hold *except* that some of the g_i's in equation (4) are replaced by constant functions. (This was still unclear to me until I read your reply to reviewer YEeM.)

---

> ### Author Response · Authors · 2023-11-21
> **Response to the Comment**
>
> Dear Reviewer 22Xe,
>
> Thank you for such a quickly replay.
>
> For Q1, we have provided an explicit definition in the latest version.
>
> For condition (iii), we have also revised it in the latest version, following a common style found in the majority of existing works that leverage results from nonlinear ICA and incorporate assumption (iii). That is,
>
> There exist $2\ell+1$ values of $\mathbf{u}$, i.e., $\mathbf{u}\_{0},\mathbf{u}\_{1},...,\mathbf{u}_{2\ell}$, such that the matrix
> $\mathbf{L} = (\boldsymbol{\eta}(\mathbf{u} = \mathbf{u}\_1)-\boldsymbol{\eta}(\mathbf{u} = \mathbf{u}\_0),..., \boldsymbol{\eta}(\mathbf{u} = \mathbf{u}\_{2\ell})-\boldsymbol{\eta}(\mathbf{u}= \mathbf{u}\_0))$ of size $2\ell \times 2\ell$ is invertible.
>
> For Corollary 3.2, please note that we do not impose any limitations on function $\lambda_{i,j}$ in equations (1)-(4). Therefore, even if equations (1)-(4) hold,  $\lambda_{i,j}(\mathbf{u})$ could be a constant. In fact, our assumption (iv) in Theorem 3.1 imposes that $\lambda_{i,j}(\mathbf{u})$ can not be a nonzero constant.

---

> ### Author Response · Authors · 2023-11-22
> **Could you please confirm if the clarification we supplied sufficiently addresses your concerns?**
>
> Dear Reviewer 22Xe,
>
> Your feedback is of immense importance to us. Could you kindly verify if the provided clarification effectively addresses your concerns, especially regarding the core concepts?
>
> Should any uncertainties persist or if additional clarification is warranted, please donot hesitate to inform us. We recognize the demands on your time and sincerely appreciate your thoughtful consideration. Your reassessment plays a pivotal role in advancing our work, and we stand ready to provide any further clarification you may require.
>
> Best regards,
> Authors

---

> ### Author Response · Authors · 2023-11-23
> **Follow-Up: Rebuttal**
>
> Dear Reviewer 22Xe,
>
> We are writing to express our sincere appreciation for the time and effort you have dedicated to reviewing my manuscript. Your valuable feedback has been instrumental in refining the quality of the work.
>
> We are currently in the process of finalizing the revisions based on your insightful comments. As the deadline for rebuttal is approaching, We wanted to reach out and ensure that we have adequately addressed any concerns you may have had.
>
> **In response to your feedback, We have focused on:**
> 1) We have extracted the definition of identifiability from Theorem 3.1 and formalized it as Definition 3.1.
> 2) We have reexpressed the assumption (iii) in Theorem 3.1.
> 3) We have taken special care to clarify that even if equations (1)-(4) are satisfied, $\lambda_{i,j}(\mathbf{u})$ could be a constant.
>
> We believe these revisions contribute to the overall clarity and robustness of the manuscript.
>
> We understand that your schedule may be busy, and we greatly appreciate the time you have already invested in the review process. However, as the deadline is imminent, we wanted to kindly remind you of the approaching date and express my eagerness to incorporate any additional suggestions or address any lingering concerns you may have.
>
> We are more than willing to provide further clarification or discuss any aspects of the manuscript in more detail. Your feedback is invaluable, and we want to ensure that the latest version of the manuscript meets the high standards.
>
> Thank you once again for your time and consideration.
>
> Best,
>
> Authors

---

### Official Review · Reviewer_KC1H · 2023-10-31

**Soundness:** 4 excellent
**Presentation:** 3 good
**Contribution:** 3 good
**Rating:** 6
**Confidence:** 4

**Summary:**

This paper proposes a novel theoretical result for causal representation learning, where the latent causal variables and structures are identified by exploiting the changes of causal influences across multiple environments. Built upon (Liu et al., 2022), the paper generalizes this previous result from linear Gaussian models to polynomial models with exponential family noise, as well as reducing the number of required environments. Identifiability results under different scenarios of changing coefficients were also discussed in detail. The paper also presents an empirical estimation method and validates it on synthetic and real-world data

**Strengths:**

- In general the writing and presentation of the relatively clear and easy to understand.

- The paper tackles an important and challenging problem of learning latent causal representations from observational data,  and makes significant theoretical contributions by extending the scope of latent causal models to nonlinear and non-Gaussian cases, and relaxing the number of required environments.
- The paper did not stop after presenting their main results, but also provides in-depth discussions on partial identifiability results which is very useful and insightful. This will help provide more guidance under more realistic scenarios in practice.

**Weaknesses:**

I appreciate your feedback. Here's an improved version of the weaknesses section:

1. **Latent structure identifiability**: While the paper makes significant strides in latent variable identifiability, it does not discuss the identifiability of the latent structure under the equivalence class. I understand that once the latent variable values are identified, the latent structure can be recovered trivially. However, given that the latent variables are only recovered up to an equivalence class, especially for partial identification case, it is unclear how would these affect the latent structure identification. This is a crucial aspect in causal representation learning, as understanding the underlying causal structure is often more important than identifying individual variables. The paper could benefit from a more thorough discussion or analysis on this aspect.

2. **Functional assumption**: The necessity and expressiveness of the polynomial assumption are not adequately addressed in the paper. While it's understood that this assumption is needed for identifiability, it would be beneficial to have a more detailed discussion on why this specific form is chosen and how expressive it can be in capturing complex causal relationships. It would also be interesting to explore if there are other forms or assumptions that could achieve similar results.

3. **Presentation**: Given that most of the assumptions have a strong graphical interpretation, the paper could benefit from adding more illustrations or visualizations to help readers better understand these assumptions and their implications.

**Questions:**

See my concerns in previous sections.

---

> ### Author Response · Authors · 2023-11-19
> **Response**
>
> **Q1: Latent structure identifiability**
>
> **R1**: Thank you for your high-level and insightful suggestions. Our primary finding, as presented in Theorem 3.1, demonstrates that latent causal variables can be identified up to permutation and scaling. Importantly, permutations do not affect the underlying latent graph structure, as they are solely related to namespace. Regarding scaling, as mentioned in our paper, the identifiability of nonlinear models with additive noise remains unaffected by scaling, as we claimed in the paragraph following the proof sketch for theorem 3.1. Consequently, the equivalence class, encompassing permutation and scaling, corresponds to the same graph structure. In terms of partial identifiability, we think that our current results only allow for the identifiability of the values of a subset of latent variables. In such cases, asserting the recovery of a portion of the true causal graph is challenging, given our belief that the unidentifiable part may introduce various possible effects on the graph structure within the identifiable portion. We argue that understanding the underlying causal structure is often more crucial than identifying individual variables. However, for certain applications, such as domain adaptation, we may be interested in splitting the latent space into an invariant space and a variant space, as domain adaptation can leverage the invariant part to predict labels, regardless of the graph structure in latent space. We have incorporated these discussions in the new version. Please see **Section A.8**  in the revised version for more details.
>
> **Q2: Functional assumption**
>
> **R2**: Thank you once again for your insightful suggestions. In general, we believe that model assumptions can be extended to a more general case than polynomial, even for non-parametric models, as long as the changes in causal influences are sufficient. However, defining what sufficient changes poses a challenge. In this work, we opted for polynomial models, considering their approximation ability and simple expression, which is beneficial for analysis and formulating the notion of sufficient changes. Simultaneously, we acknowledge the limitations of this choice, particularly with the presence of high-order terms in polynomials, making optimization highly challenging. Please see **Section A.9** in the revised version for more details.
>
> **Q3: Undestanding the assumptions**
>
> **R3**: Thank you once again. We also recognize the significance of facilitating a better understanding of these assumptions for our readers. Consequently, we have augmented our presentation by including a dedicated in **Section A. 10**  in the revised version to provide detailed insights into these assumptions. For example, two of the main assumptions (iii) and (iv) in Theorem 3.1. Assumption (iii) stems from nonlinear ICA, essentially requiring sufficient changes in latent noise variables to facilitate their identification. Assumption (iv) is derived from the work by Liu et al. (2022), aiming to constrain the function class of lambda to prevent a specific case, e.g., lambda(u) = lambda(u) + b, where b is a constant. In this scenario, the invariant b leads to unidentifiability, as discussed in Corollary 3.3.

---

### Official Review · Reviewer_UsXb · 2023-11-01

**Soundness:** 2 fair
**Presentation:** 4 excellent
**Contribution:** 3 good
**Rating:** 6
**Confidence:** 4

**Summary:**

The papers deal with the identification of latent causal models from high-dimensional observations and provide results for the case of polynomial causal relationships, which extends over the prior results that were limited to linear causal relationships along with gaussian noise. They utilize the commonly used assumption of an exponential family distribution conditioned on surrogate variables for the noise variables, which helps them identify the noise variables up to permutation and scaling. Then they show how the identification guarantees on the noise variables can be used to get identification guarantees on the latent variables themselves using the assumption of polynomial functional relationships that show variation with the surrogate variable, similar to recent works leveraging interventional data for latent identification.

**Strengths:**

* The paper is well-written with a clear description of the various assumptions needed for the theoretical results, along with an extremely good proof sketch! Further, the discussion on prior work is quite easy to follow and places the contributions of the work in a good manner as compared to them.

* The identification result provided in the paper uses fairly standard assumptions in the literature, but the application for the task of learning the latent causal relationships (not just the latent causal variables) is novel, and also somewhat significant as it is not limited only to the case of linear gaussian latent causal model.

* The proposed approach is principled and the claims made in the paper are supported by experimentation over various benchmarks, however, the scale of the experiments is limited (refer to the Weaknesses section below for more details).

**Weaknesses:**

* My major concern with the work is regarding the technical contribution and significance of the proposed identification result. The proof uses common ideas of exponential family distribution [1] to obtain identification guarantees up to permutation and scaling for the noise variables. Further, the idea of using the changing distribution of latent variables identified up to polynomial mixing can be used to obtain permutation and scaling identification guarantees has been explored in prior works [2]. While I agree that the application of these ideas to the task of learning latent causal relationships is not trivial (as noted in the Strengths section above), I am not convinced by the significance of the proposed identification result. In terms of only the latent identification (not recovering causal relationships), the prior works that leverage interventional data can already provide permutation and scaling guarantees with much more general latent structures. For the task of learning the latent causal relationships, the permutation indeterminacy in the learned latents would pose a huge challenge which the authors circumvented by assuming the correct topological order. This makes it difficult for me to assess the importance of the proposed identification result.

* While I appreciate the experimentation over several benchmarks, the scale (dimension of the latent variables) of the experiments is too small, with the maximum dimension of the latent space being 5. It would be nice to see the results with the proposed approach for larger latent dimensions ($d= 10$ or $d=20$ at least) for the synthetic benchmark.

References:

[1] Aapo Hyvarinen, Hiroaki Sasaki, and Richard Turner. Nonlinear ica using auxiliary variables and
generalized contrastive learning. In The 22nd International Conference on Artificial Intelligence
and Statistics, pp. 859–868. PMLR, 2019.

[2] Kartik Ahuja, Divyat Mahajan, Yixin Wang, and Yoshua Bengio. Interventional causal representa-
tion learning. In International Conference on Machine Learning, pp. 372–407. PMLR, 2023.

**Questions:**

I would mostly like a follow-up discussion on the concerns I raised in the Weaknesses section above. There are some other minor question mentioned below:

* Is there any particular reason why the authors did not compare against several latent identification works that leverage interventional data? That would be a better choice of baselines as compared to Beta-VAE, which does not explicitly utilize the variation of latent variables with auxiliary variables.

* It would be good to plot the F1 score along with SHD for the performance of latent causal discovery as well. The authors could maybe set a threshold to determine the latent causal structure from the functional relationships and compare that against the true latent causal graph.

---

> ### Author Response · Authors · 2023-11-19
> **Response**
>
> **Q1: The proof uses common ideas of exponential family distribution [1] to obtain identification guarantees up to permutation and scaling for the noise variables.**
>
> **R1**: It appears there might be confusion. We main result, Theorem 3.1, is using the result from [2], not [1], to obtain identifiability up to permutation and scaling for the noise variables.
>
> [1] Aapo Hyvarinen, Hiroaki Sasaki, and Richard Turner. Nonlinear ica using auxiliary variables and generalized contrastive learning. In The 22nd International Conference on Artificial Intelligence and Statistics, pp. 859–868. PMLR, 2019.
>
> [2] Sorrenson, Peter, Carsten Rother, and Ullrich Kothe. "Disentanglement by nonlinear ica with general incompressible flow networks." arXiv preprint arXiv: 2001.04872 (2020).
>
> Please be aware of a substantial disparity between the identifiability of latent noise variables n and latent causal variables z. Notably, assuming an absence of all edges among z makes it trivial to apply the results of nonlinear ICA. However, in real-world applications, latent graph structures can be arbitrary. Consequently, the solution space for identifying z is notably more expansive than that for identifying n. This disparity is critical, especially given the complexity introduced by arbitrary graph structures. Essentially, many existing works focusing on the changes of causal influences, e.g., hard intervention or soft intervention, are dedicated to addressing this pronounced gap.
>
> **Q2: Further, the idea of using the changing distribution of latent variables identified up to polynomial mixing can be used to obtain permutation and scaling identification guarantees has been explored in prior works [2].**
>
> **R2**: It is essential to highlight that [2] achieves the recovery of latent causal variables identified up to polynomial mixing by assuming a polynomial mapping from latent causal variables z to observed x. In contrast, our approach employs a more general mapping f, as defined in Eq. (3). Consequently, in our setting, the task of recovering latent causal variables identified up to polynomial mixing becomes more challenging due to the increased flexibility and complexity introduced by the general mapping function f.
>
> Most importantly, the distinction between this work and [2] is substantial. [2] utilizes hard interventions to identify latent causal variables, while this work employs soft interventions for the same purpose. To elucidate this significant difference, let us consider a fundamental question: how can hard interventions be practically applied to latent causal variables?
>
> We posit that conducting hard interventions on latent space is almost impractical due to the unobservable nature of latent causal variables. Consequently, hard interventions should be understood as modeling changes generated by self-initiated behavior within causal systems across environments. However, the self-initiated changes in causal systems across environments can be arbitrary. In this context, hard interventions in [2] only allow model specific types of changes, while this work, involving soft interventions, can model a broader range of possible changes. This versatility may prove more attainable for latent variables than the constrained nature of hard interventions.
>
> **Q3: ...In terms of only the latent identification (not recovering causal relationships)...**
>
> **R3**: Please take note that our identifiability results, elucidated in Theorem 3.1, inherently signify the identifiability of the causal graph structure, as explicitly stated in the paragraph following the proof sketch for Theorem 3.1. This assertion is grounded in the inherent identifiability of nonlinear models with additive noise. Please see the detailed clarification provided in the corresponding paragraph for a more thorough and comprehensive understanding of this crucial aspect. Furthermore, it is important to highlight that our results are not contingent upon any assumptions regarding latent structures. This underscores the flexibility of our framework, allowing for the consideration of arbitrary structures.

---

> > ### Author Response · Authors · 2023-11-19
> > **Response**
> >
> > **Q4: the permutation indeterminacy in the learned latents would pose a huge challenge which the authors circumvented by assuming the correct topological order.**
> >
> > **R4**: We would like to clarify that we do not circumvent permutation; rather, we strategically leverage it to overcome traditional Directed Acyclic Graph (DAG) constraints, as proposed by Zheng et al. (2018). To illustrate this concept, consider a scenario with only two latent causal variables, namely size (z1) and color (z2) of an object, with z1 causing z2. Upon obtaining two recovered latent causal variables, z'1 and z'2, permutation indeterminacy arises, allowing z'1 to correspond to either the size (z1) or the color (z2). This flexibility empowers us to pre-define a causal order in the inference model, such as z'1 causing z'2, without specifying semantic information (size or color) for the nodes. Consequently, the inference model establishes a causal relationship between z'1 and z'2, prompting them to learn the genuine latent semantic information. In other words, the inference model compels z'1 (or z'2) to acquire size (or color) information, effectively sidestepping DAG constraints. This strategic use of permutation indeterminacy enhances the adaptability of our approach. For a more comprehensive and detailed explanation, please refer to Section 3.4 of the work by Liu et al. (2022), where the authors provide a thorough explanation elucidating the rationale behind the predefinition of a causal order in inference models.
> >
> > **Q5: I appreciate the experimentation over several benchmarks, the scale (dimension of the latent variables) of the experiments is too small.**
> >
> > **R5**: Currently, causal representation learning is in its initial stage, with a predominant focus on pivotal steps like identifiability in most existing works. The optimization problem in latent space is acknowledged as a significant challenge. Typically, only an approximation of the true latent variables can be obtained, and even a slight error in this approximation can lead to diverse learned causal graphs. This is one reason why many existing works opt for experiments in settings with small dimensions of latent variables. In typical cases, the value of d is less than 10. We anticipate that identifiability results will converge in the coming years, paving the way for more researchers to address the optimization challenges that follow. We will expand our experiments to include latent variables with larger dimensions and offer a more in-depth discussion in final version, even if the results may not be as successful. Thanks for your suggestion.
> >
> > **Q6: Is there any particular reason why the authors did not compare against several latent identification works that leverage interventional data?**
> >
> > **Response**:  The fundamental issue lies in the lack of alignment in assumptions among works employing interventions. For example, there is a divergence in the nature of interventions, with some works assuming hard interventions and others assuming soft interventions. Additionally, discrepancies extend to the choice of models, where some studies assume linearity within latent spaces, while others opt for nonlinear models. There are also variations in the requirement for paired interventional data, whereas this study focuses on unpaired data. These misalignments present significant hurdles for conducting fair and meaningful comparisons, rendering it challenging for most existing works, including this one, to offer comprehensive and meaningful comparisons. Further, we anticipate that the assumptions for identifiability will converge in the coming years, paving the way for fair comparisons. We think this convergence promising, as the utilization of both hard and soft interventions is essential for modeling the changes in causal influences.
> >
> > **Q7: It would be good to plot the F1 score**
> >
> > **R7**: We appreciate your input and will add it for the final version. However, given the constraints of the rebuttal time frame, we must prioritize addressing more complex reviews. Your understanding is greatly appreciated.
> >
> > --------------------------
> > Dear reviewer UsXb,
> >
> > Thank you sincerely for investing your time and consideration into this evaluation process. Our aim in providing this clarification is to furnish valuable insights that address your concerns and contribute to a more nuanced understanding of the contributions and merits of our research. Your thoughtful re-evaluation holds immense significance for us, and we sincerely hope that our clarification proves instrumental in facilitating a comprehensive and constructive reassessment on your part.

---

> > > ### Comment · Reviewer_UsXb · 2023-11-22
> > > **Good rebuttal!**
> > >
> > > Thanks for your detailed response, especially regarding the permutation indeterminacy affecting the structure learning. Indeed I had misunderstood earlier and I have a better understanding of this part now. I would still encourage the authors to incorporate the suggestions regarding the experiments. I have increased my score accordingly and will engage in discussion with reviewers further.

---

> ### Author Response · Authors · 2023-11-22
>
> We will add experiments in the final version! We sincerely appreciate the time and effort you dedicated to reviewing our work.

---

### Official Review · Reviewer_kugU · 2023-11-01

**Soundness:** 3 good
**Presentation:** 2 fair
**Contribution:** 1 poor
**Rating:** 3
**Confidence:** 3

**Summary:**

The present manuscript is concerned with the extension of Liu et al. 2022's identifiability result to non-linear models with more general noise terms. The authors present as core contribution a theorem that allows for polynomial models and exponential family noise. A brief discussion on the violation of change of the exogenous terms is presented. Finally, an empirical section on non- and synthetic data sets.

**Strengths:**

IMHO the paper's noteworthy strengths are limited to their idea rather than the execution, therefore, they should be considered (where applicable) as counterfactuals for the moment. Said "potential" strengths are considered one-by-one in the following list (the list is ordered in correspondence to the paper presentation):
* Causal Representation Learning is an exciting and challenging avenue of research, naturally, extending prior results, especially the generalization sort of results, through precise characterization of key necessities is of great value. The authors place an effort in getting prior results to work for larger model classes such as polynomials.
* An effort of presenting a self-enclosed treatise, covering both theory and empirics.
* A discussion of partiality, as arguably most of the time in practice we work with approximations.
* Although this should IMO not count as "strength", given that a large portions of papers within the community don't abide by this, but the fact that the authors present a clear and transparent communication of their assumptions.

**Weaknesses:**

The paper suffers from several disadvantages, ranging in importance from minor to more fundamental. The more fundamental ones are of major content/technical concern, and given that this review takes a content-centric approach, they weigh the most for the low scoring. The following list - again one-by-one - aims to provide specific pointers with improvement suggestions if applicable (please note, the list is unordered):
* Liu et al. 2022 plays a central role in this work, since this work poses a direct successor in the form of an incremental improvement. While an incremental improvement can be of substantial nature, IMHO this work poses a Corollary-type of discussion over Liu et al. Furthermore, the presentation (especially in direct comparison to Liu et al.) does not provide any further justice to the present manuscript's proposed contribution. There is an abundance of repetition of the form "we've extended it now" accompanied by a lack of how alleged extension was achieved. Consulting the appendix did not resolve this major concern.
* Kronecker product definition is not clear, not the usual definition e.g. $(a_1, a_2) \times B = (a_1\cdot B, a_2\cdot B)$.
* Although I'm confident in being able to guess what the authors refer to in their convoluted discussion on the data-generating process, it really does not help the fact that all shades of exogenous/latent terms (z, u, n) are not being explored thoroughly from first principles. Putting myself in the shoes of a reader who would be unfimiliar with the literature, I'd be tempted to expect that this portion would lead to major confusion.
* Unusual, and IMHO absurd, number of assumptions paired with severity in some cases (e.g. bijectivity of $\mathbf{f}$ is exploited oftentimes and arguably standard within the literature, although this should be questioned as well at some point, but existence of a suitable $\mathbf{L}$ seems more tricky). This point shines especially in considerations of claims of appraisal made by the authors, such as "[...] significantly bridg[ing] the divide between foundational theory and practical applications", that seem to overestimate the presented contribution(s) and this, with all due to respect, in a debatably arrogant manner.
* Corollary 3.2. seems to carry no value when considering that the whole paradigm of change of causal influence is based on the complete change assumption.

Finally, following list provides concrete recommendations/pointers as to how the work might be improved such that its potential value for the community can be recognized/achieved. Again, this review is intended, respectfully, as constructive criticism and by no means as an attack to the authors. Sincerely, I hope that some of the below aspects can help:
* Given the heavy reliance on Liu et al. 2022, the authors should consider actually saying what "transitivity" is within their introductory section and avoid using formulations that Liu et al. use in different contexts (e.g. "absorbs").
* Please consider rewriting the introduction s.t. redundancy with Section 2 is being reduced. The extra space that is won through this is valuable presentation space, which avoids having to put otherwise more important results into the appendix.
* Please consider rephrasing Fig.1 caption as not simply being a copy of the introductory text description of the Figure.
* Please consider removing the pre-text at the beginning of Section 3, in favor of relevant discussions, as it arguably provides no additional value given all the text before (Abstract and Sec.1,2)
* Please consider improving accessibility to the Figures by resolving the issues on the following two dimensions: (1) label/legend and general font sizes but also (2) proportions e.g. the choice of content being presented (e.g. Figure 6 and 7 do not convey valuable information w.r.t. this aspect).

**Questions:**

TL;DR: No questions.

Even though I've checked the appendix, I've not done so in utmost detail, still, I'm confident in being able to guess what the authors actually mean or refer to in situations of uncertainty, therefore, no questions are derived on that end. Furthermore, the lack of discussions on the actual key insights, for making the transition to polynomials & the exponential family work, does not allow me to raise any further questions.

---

> ### Author Response · Authors · 2023-11-19
> **Response**
>
> **Q1: an incremental improvement, a lack of how alleged extension was achieved**
>
> **Response**: We respectfully disagree with that. Our contributions, highlighted throughout the abstract, introduction, and theoretical results, encompass the following advancements:
>
> 1) The dependencies among latent variables, previously assumed to be linear in Liu's work, are demonstrated to be polynomial in our research.
> 2) Contrary to the Gaussian assumption in Liu's work, we establish that the noise associated with latent variables follows an exponential distribution.
> 3) In comparison to Liu's prior work, our proof requires a significantly reduced number of environments (2l+1 environments), a substantial improvement from the previous quadratic requirement in Liu's work.
> 4) We also provide proofs for partial (un)identifiability, specifically addressing situations where the assumption of a complete change in causal influences is violated.
> 5) Introducing an estimation method, we validate its efficacy on both synthetic and real-world datasets.
>
>
> All theoretical contributions are rigorous justified by the Theorem 3.1, Corollary 3.2, Corollary 3.3, and their respective proofs. Further, experimental results serve to affirm not only the soundness of our theoretical contributions but also the practical efficacy of the proposed estimation method. Finally, to understand the motivation behind these theoretical advancements, our Introduction sheds light on the fact that LIU's work primarily focuses on identifying latent causal variables from the perspective of observed data. In contrast, our research is propelled by the realization that we can significantly narrow down the solution space of latent causal variables by considering the perspective of latent noise variables. This new insight motivates us to propose more general models than LIU's work.
>
> **Q2: Kronecker product definition is not clear, not the usual definition.**
>
> **Response**: We deviate from the conventional definition by employing the notation ${\bar \otimes}$ in place of the usual ${\otimes}$ (usually denote the Kronecker product) to denote the operator. Additionally, we explicitly elucidate that the operator, denoted as ${\bar \otimes}$, signifies the Kronecker product with all distinct entries.
>
> **Q3: Reviewer kugU, Q3: Unusual, and IMHO absurd, number of assumptions**
>
> **Response**: We respectfully disagree with this point. Many of the assumptions, encompassing i to iii, stem from the realm of nonlinear ICA. It is important to acknowledge that concerns related to objectivity and the existence of $\mathbf{L}$ pertain to assumptions ii and iii. Over recent years, nonlinear ICA has found applications in diverse real-world scenarios, such as domain adaptation [1] and image generation and translation [2], domain generalization [3]. Consequently, we assert that these three assumptions have been substantiated through practical applications.
>
> The remaining assumptions, including model assumptions and assumption (iv), revolve around the changes of causal influences. These changes are firmly grounded in the prevalent occurrence of distribution shifts in real-world applications, as elucidated in the first paragraph, far from being speculative. In contrast to many prior works that necessitate hard interventions to model the potential changes above, our approach employs soft interventions, enabling the modeling of a broader spectrum of conceivable alterations.
>
> [1] Kong, Lingjing, et al. "Partial Identifiability for Domain Adaptation." ICML 2022.
>
> [2] Xie, Shaoan, et al. "Multi-domain image generation and translation with identifiability guarantees." ICLR 2022.
>
> [3] Wang, Xinyi, et al. "Causal balancing for domain generalization." arXiv preprint arXiv:2206.05263 (2022)
>
> **Q4**: all shades of exogenous/latent terms (z, u, n) are not being explored thoroughly from first principles.
>
> **Response**:  Again, I respectfully dissent from this point. Within the causal representation community, there is a widely accepted understanding that observed data is generated by latent causal variables z. It is also well-established that each latent variable zi corresponds to a specific exogenous factor ni in causal systems. Furthermore, we assert that the variable (u) serves as a surrogate, capturing changes within the system. These conceptualizations align with established conventions in the community, and we contend that they are readily embraced by the community members.

---

> > ### Author Response · Authors · 2023-11-19
> > **Response**
> >
> > **Q5: Corollary 3.2. seems to carry no value when considering that the whole paradigm of change of causal influence is based on the complete change assumption.**
> >
> > **Response**: There may be a misunderstanding of the Corollary 3.2 in the review. The Corollary aims to explore the necessity of the assumption that requiring all coefficients within polynomial models to change for complete identifiability, without relying on any additional assumptions. Exploring the necessity of the assumption is important for robustness of identifiability results, generalizability, and so no.
> >
> > **Q6: The suggestions for presentation.**
> >
> > **Response**: We sincerely appreciate these valuable suggestions regarding our presentation and will thoughtfully consider their implementation.

---

> ### Author Response · Authors · 2023-11-22
> **Could you kindly verify if the provided clarification addresses your concerns?**
>
> Dear Reviewer kugU,
>
> We greatly appreciate your feedback. Could you kindly verify if the provided clarification addresses your concerns, particularly regarding contributions and assumptions?
>
> We wish to highlight our contributions once again. The challenge of identifying causal representations in learning is particularly formidable. Current approaches, which mainly focus on hard interventions through changes in causal influences (distribution shifts), are limited. We contend that implementing hard interventions in the latent space is nearly impractical due to the unobservable nature of latent causal variables.As a consequence, hard interventions should be interpreted as modeling changes generated by self-initiated behavior within causal systems across diverse environments. However, self-initiated changes in causal systems can be arbitrary. In this context, hard interventions only permit model-specific changes, whereas our approach, involving soft interventions, has the potential to model a broader range of possible changes. To our knowledge, the work of Liu et al. 2022 is the sole instance utilizing soft interventions, albeit confined to special linear Gaussian models. In contrast, our work advances this by considering nonlinear models with exponential family noise. This transition from linear to nonlinear, from Gaussian noise to exponential family noise, is a significant step forward. Importantly, it also mitigates the requirement for a high number of environments.
>
> If there are lingering uncertainties or if additional clarification is necessary, please don't hesitate to notify us. We understand the time constraints you face and truly appreciate your thoughtful consideration. Your reassessment plays a crucial role in advancing our work, and we stand prepared to offer any further clarification needed.
>
> Best regards,
> Authors

---

> > ### Author Response · Authors · 2023-11-23
> > **Follow-up: Rebuttal**
> >
> > Dear Reviewer kugU,
> >
> > We wish to express our sincere gratitude for the dedicated time and effort you have committed to reviewing our manuscript. Your invaluable feedback has played a pivotal role in enhancing the overall quality of our work.
> >
> > Currently, we are in the process of finalizing revisions based on your insightful comments. With the rebuttal deadline approaching, we are reaching out to ensure that we have thoroughly addressed any concerns you may have raised.
> >
> > **In consideration of your feedback, our response above has emphasized key areas related to contributions and assumptions.**
> >
> > Recognizing the demands on your schedule, we appreciate the time you have already invested in the review process. However, as the deadline is imminent, we are more than willing to provide further clarification or engage in detailed discussions on any aspects of the manuscript. Your feedback is invaluable.
> >
> > Thank you once again for your time and consideration.
> >
> > Best regards,
> >
> > Authors

---

### Author Response · Authors · 2023-11-19
**General Response**

We extend our sincere appreciation to the reviewers for their invaluable feedback. It is gratifying to receive positive remarks across various dimensions:

**Contribution**:

" is novel, and also somewhat significant," --  Reviewer UsXb

"makes significant theoretical contributions" -- Reviewer KC1H,

"I believe the extension proposed here is a significant improvement compared to previous work." -- Reviewer 22Xe,

" the paper makes a few novel contributions in this challenging domain" -- Reviewer YEeM

**Writing**:

"The paper is well-written", "an extremely good proof sketch", "the discussion on prior work is quite easy to follow" -- Reviewer UsXb

"In general the writing and presentation of the relatively clear and easy to understand", "" -- Reviewer KC1H

"The writing is overall of a good quality." -- Reviewer 22Xe

"The paper is very well written", "The background is clearly explained and sufficient", "Sketches of proofs are given for the theorems, which nicely give intuition", "Mathematical parts (model, theorems, training) are rigorously presented",
"Overall, the presentation makes a nice trade-off between accessibility and rigorousness" -- Reviewer YEeM

**Experiments**:

"The experiments are reasonably thorough and demonstrate and complement the theoretical developments in an intuitive way." -- Reviewer YEeM


In response to specific concerns raised by each reviewer:

Reviewer kugU raised main concerns about contribution and assumptions, we have provided further clarification.

Reviewer UsXb raised main technical contribution and experiments, we have provided further clarification.

Reviewer KC1H suggested an improved version of the weaknesses, including structure identifiability, polynomial assumptions, and understanding assumptions, we have added further dicussions in sections A.8, A.9, and A.11.

Reviewer 22Xe raised a concern on the presentation of core concepts, which are addressed by further explanation.

Reviewer YEeM raised some technical details, which are addressed by further explanation and adding a new section A. 11.

We have answered all questions (see more details in the individual responses).

We sincerely thank all reviewers for their thorough evaluations. We are open to further discussion and welcome any additional feedback.

---

### Meta-Review · Area_Chair_d8gz · 2023-12-06

**Metareview:**

The paper extends identifiability exploration to encompass nonlinear causal relationships represented by polynomial models and general noise distributions within the exponential family. It also explores the necessity of change.

pros:
+ The paper makes contributions in the challenging domain of latent causal modeling.
+ The experiments are reasonably thorough and complement the theoretical developments.

cons:
+ The paper relies heavily on Liu et al;
+ Empirical experiments lack real-world relevance and do not cover difficult cases.
+ Lack of thorough discussion on latent structure identifiability and functional assumption.

**Justification For Why Not Higher Score:**

Lack of sufficient real world experiments; lack of sufficient discussions on identifiability and assumptions

**Justification For Why Not Lower Score:**

reasonably thorough contributions to a challenging problem

---

### Decision · Program_Chairs · 2024-01-16

Accept (poster)